# Human IgG Fc-engineering for enhanced plasma half-life, mucosal distribution and killing of cancer cells and bacteria

Stian Foss[1,2,3], Siri A. Sakya [1,2,3,15], Leire Aguinagalde[4,15], Marta Lustig[5,15], Jutamas Shaughnessy [6,15], Ana Rita Cruz[4,15], Lisette Scheepmaker[4,15], Line Mathiesen [7], Fulgencio Ruso-Julve [1,2,3], Aina Karen Anthi [1,2,3], Torleif Tollefsrud Gjølberg [1,2,3], Simone Mester[1,2,3], Malin Bern[1,2,3], Mitchell Evers[8], Diane B. Bratlie[9], Terje E. Michaelsen[9,10], Tilman Schlothauer[11], Devin Sok[12], Jayanta Bhattacharya[13], Jeanette Leusen[8], Thomas Valerius [5], Sanjay Ram [6], Suzan H. M. Rooijakkers [4], Inger Sandlie[14] & Jan Terje Andersen [1,2,3] ✉

Monoclonal IgG antibodies constitute the fastest growing class of therapeutics. Thus, there is an intense interest to design more potent antibody formats, where long plasma half-life is a commercially competitive differentiator affecting dosing, frequency of administration and thereby potentially patient compliance. Here, we report on an Fc-engineered variant with three amino acid substitutions Q311R/M428E/N434W (REW), that enhances plasma half-life and mucosal distribution, as well as allows for needle-free delivery across respiratory epithelial barriers in human FcRn transgenic mice. In addition, the Fc-engineered variant improves on-target complement-mediated killing of cancer cells as well as both gram-positive and gram-negative bacteria. Hence, this versatile Fc technology should be broadly applicable in antibody design aiming for long-acting prophylactic or therapeutic interventions.

More than 100 monoclonal IgG antibodies have been approved for treatment of a range of diseases, including cancer, chronic inflammation, diabetes, migraine, and cardiovascular disorders[1,2]. This has spurred an extensive interest in engineering of antibody formats with improved efficacy for both therapeutic and prophylactic use[3]. While most antibodies are directed against hematological and solid tumors, development of antibodies to prevent or treat infectious and neglected diseases is a growing field, as exemplified by the availability of licensed antibodies against SARS-CoV-2 to slow COVID-19 progression[4,5]. Tailored

[1]Department of Immunology, Oslo University Hospital, Rikshospitalet, Oslo, Norway. [2]Institute of Clinical Medicine, Department of Pharmacology, University of Oslo, Oslo, Norway. [3]Precision Immunotherapy Alliance (PRIMA), University of Oslo, Oslo, Norway. [4]Medical Microbiology, University Medical Center Utrecht, Utrecht University, Utrecht, the Netherlands. [5]Section for Stem Cell Transplantation and Immunotherapy, Department of Medicine II, Christian-Albrechts University Kiel and University Medical Center Schleswig-Holstein, Kiel, Germany. [6]Department of Medicine, Division of Infectious Diseases and Immunology, University of Massachusetts Chan Medical School, Worcester, MA, USA. [7]Department of Public Health, Faculty of Health Sciences, University of Copenhagen, Copenhagen, Denmark. [8]Center for Translational Immunology, University Medical Center Utrecht, Utrecht, the Netherlands. [9]Infection Immunology, Norwegian Institute of Public Health, Oslo, Norway. [10]Department of Chemical Pharmacy, School of Pharmacy, University of Oslo, Oslo, Norway. [11]Roche Pharma Research and Early Development (pRED), Roche Innovation Center Munich, Munich, Germany. [12]International AIDS Vaccine Initiative (IAVI), New York, NY, USA. [13]Antibody Translational Research Program, Translational Health Science & Technology Institute, NCR Biotech Science Cluster, Faridabad, India. [14]Department of Biosciences, University of Oslo, Oslo, Norway. [15]These authors contributed equally: Siri A. Sakya, Leire Aguinagalde, Marta Lustig, Jutamas Shaughnessy, Ana Rita Cruz, Lisette Scheepmaker. ✉e-mail: j.t.andersen@medisin.uio.no

antibodies may also become important treatment options against the growing threat from antimicrobial resistant (AMR) bacteria[6,7].

However, further development of antibodies requires fine-tuned engineering that secures specific modes of action without causing severe side effects. For instance, antibodies targeting cancer cells or bacteria will benefit from features that specifically enhance target eradication[8]. In addition, engineering that prolongs exposure time, systemically, but also at mucosal sites, and allows antibodies to reach the site of action at high concentrations is favorable, as it may extend dosing intervals[9–11]. Such improvements may improve patient convenience, treatment adherence and reduce costs for the healthcare system. Enhanced bioavailability has implications for treatment of life-long chronic diseases and prophylactic treatments.

Most approved antibodies are built on full-length IgG1, and the constant IgG1 Fc region is also used as a fusion partner to extend the plasma half-life of fused therapeutic modalities[12]. While natural IgG1 antibodies have a half-life of 3 weeks on average, individual monoclonal IgG1s have strikingly different half-lives, ranging from 6–32 days, while Fc-fusions mostly have shorter half-lives than IgG1[3,13]. Therefore, engineering for optimal engagement of the neonatal Fc receptor (FcRn) is crucial, as binding rescues IgG from intracellular degradation[14]. This occurs in acidic endosomes following fluid-phase pinocytosis and recycling of the FcRn:IgG complex back to the cell surface, where IgG is released into circulation[15–17]. In addition, FcRn mediates transcytosis of IgG across mucosal epithelial barriers and the human placenta[18–20]. Both recycling and transcytosis rely on strict pH dependent Fc binding to FcRn, with binding at acidic (pH 6.5–5.5) and release or no binding at neutral pH[21,22]. This insight has guided Fc-engineering of IgG variants with improved pH-dependent FcRn binding, resulting in antibodies with extended plasma half-life in human FcRn transgenic mice, non-human primates (NHPs) as well as humans[10,23–25]. Importantly, increased persistence has been shown to improve anti-tumor activity, mucosal distribution, and prophylactic activity upon challenge with for instance human immunodeficiency virus (HIV) and SARS-CoV-2[9,23,24,26]. However, such Fc-engineering may also negatively influence effector functions, such as antibody-dependent cellular cytotoxicity (ADCC), antibody-dependent cellular phagocytosis (ADCP) and complement-dependent cytotoxicity (CDC)[9,27]. Hence, there is a need for fine-tuned engineering strategies that not only improve pharmacokinetic properties, but also enhance specific effector functions with a minimum of amino acid substitutions to reduce the risk of immunogenicity.

Here, we describe an Fc-engineering approach, based on three amino acid substitutions (Q311R/M428E/N434W) (REW), which improves pH-dependent human FcRn binding. We demonstrate prolonged plasma half-life and improved biodistribution for full-length IgG antibodies and Fc-based formats in human FcRn-expressing mice, allowing for both invasive and non-invasive delivery. Furthermore, the triple substitutions considerably potentiate complement-mediated killing or phagocytosis of not only cancer cells but also gram-positive and gram-negative bacteria. Importantly, this dual engineering approach results in on-target and IgG subclass specific effector molecule engagement, giving rise to a toolbox that can guide design of IgG antibodies with tailored mode-of-action towards a specific target and indication.

## Results

### Engineering of IgG1 for prolonged plasma half-life

By scrutinizing a human FcRn:IgG1 Fc co-crystal structure[21], we identified three Fc amino acid residues that were substituted with the rationale to improve pH-dependent receptor binding, namely Q311R, M428E and N434W (REW) (Fig. 1a, Fig. S1, SI Text 1). Introduction of REW into human IgG1, specific for the hapten 4-hydroxy-3-iodo-5-nitrophenylacetyl (NIP), did not reduce production rate, as the amounts of secreted protein were equal to that of wild-type (WT) IgG1 in a

serum-free Expi293 expression system (Fig. S2, SI Text 2). Importantly, the REW substitutions did neither affect thermal stability, aggregation, or the composition of N297-linked *N*-glycan structures (Fig. S2, SI Text 2, Table S1).

Next, we demonstrated that IgG1-REW bound more strongly to human FcRn than WT IgG1 at acidic pH in ELISA, while no binding was detected at neutral pH (Fig. 1b–d). The FcRn negative IgG1 variant H310A did not bind at either pH condition. Using surface plasmon resonance (SPR), increased and reversible receptor binding was demonstrated at pH 6.0, resulting in a 20-fold lower $K_D$ than WT (Fig. 1e–g, Table S2). Next, to mimic a cellular context, where the antibodies bind the receptor before being released through an endosomal pH-gradient, the antibodies were applied at acidic pH (6.0) to a column coupled with human FcRn followed by elution as a function of a gradual increase in pH toward 8.8[28]. The results revealed that REW eluted later (pH 7.58) than the WT (pH 6.95), which supports a competitive advantage for receptor engagement (Fig. S3a, b, Table S3).

In line with this, the REW substitutions were shown to enhance IgG1 rescue from intracellular degradation in a human endothelial cell-based recycling assay (HERA) (Fig. 1h–k), resulting in a 7-fold higher so-called HERA score for REW than the WT antibody (Fig. 1l), predictive of extended plasma half-life[29]. To confirm this, the antibodies were given intravenously (i.v.) to human FcRn transgenic mice (Tg32), preloaded with pooled human IgG (IVIg) to introduce natural competition for receptor binding (Fig. 1m), which is important as these mice have low levels of endogenous mouse IgG due to lack of binding to the human receptor[30–32]. This revealed that REW-containing anti-NIP IgG1 has a plasma half-life of 2 weeks (14.2 days), about 2-fold longer than WT IgG1 with the same specificity (7.8 days) (Fig. 1n). Similarly, extended plasma half-life was measured when REW was introduced into a recently characterized anti-SARS-CoV-2 neutralizing antibody (THSC20.HVTR04) (mAb4)[33] (10.7 days) compared to WT (5.9 days) (Fig. 1o). This resulted in 2 to 3-fold higher concentration of the REW-containing variants in plasma over time (Fig. S3c–d). By fitting the plasma concentrations to a non-compartmental PK model[34], we found that the IgG1-REW variants were present at higher concentrations in plasma over time (AUC) compared to WT, had slower clearance, longer mean residence time and lower volume of distribution at steady state (Table S4). Notably, the levels of total IgG, and thereby the competitive pressure on FcRn, were comparable over time between the WT and REW groups, which means that REW did not influence the IVIg levels differently than WT (Fig. S3e, f). In addition, when the same experiment was performed without pre-loading with IVIg, the anti-NIP IgG1 WT and REW variants showed similar prolonged half-lives (Fig. S3g), highlighting the important of in vivo competition.

To assess if the injected anti-NIP IgG1s triggered immunogenicity under the conditions tested, mouse anti-human IgG responses were measured by ELISA, which revealed only a minor response in mice receiving mAb4-WT but not in animals that received NIP-WT, NIP-REW or mAb4-REW when compared to pre-bleed samples (Fig. S3h–k). Extended plasma half-life of REW compared to WT was dependent on human FcRn, as the REW substitutions disrupted pH-dependent binding to mouse FcRn, and resulted in about 2-fold shorter plasma half-life in conventional mice (Fig. S3l–o), in line with data for other Fc-engineering strategies[29,35,36].

### REW enhances lung localization, transcytosis and transplacental transport

To assess biodistribution, bronchoalveolar lavage fluid (BALF) from Tg32 mice i.v. administered with anti-NIP IgG1 WT and REW were collected at the 23-day endpoint of the above plasma half-life study (Fig. 2a). Quantification of antibody in the samples revealed a 2-fold higher bioavailability of REW in the lungs (Fig. 2b). Calculating the BALF/plasma ratio revealed non-significant differences (Fig. 2c),

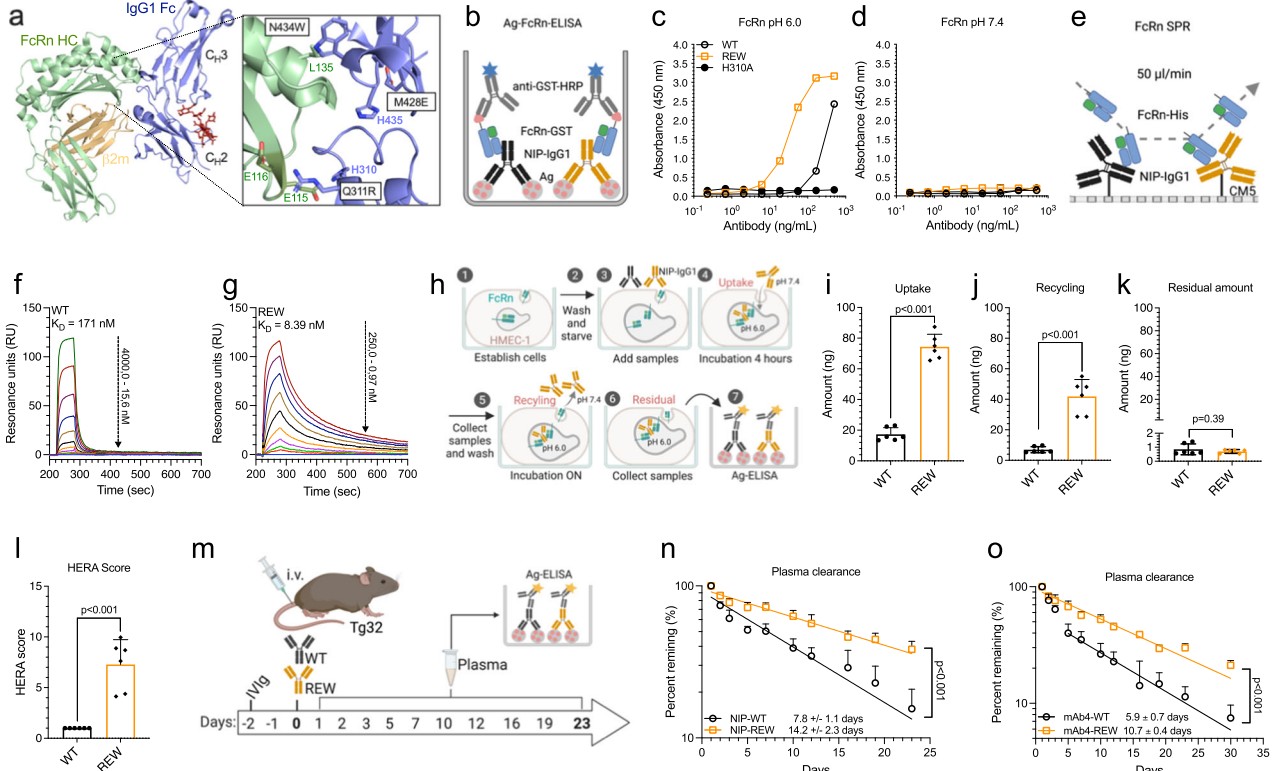

**Fig. 1 | Fc-engineered IgG1 with improved human FcRn binding and extended plasma half-life. a** The solved co-crystal structure of truncated recombinant human FcRn (green) in complex with IgG1 Fc (blue). The REW amino acid substitutions (Q311R/M428E/N434W) in the Fc are indicated. The Fc residues H435 and H310, required for pH dependent binding, as well as the FcRn residues E115, E116, and L135 are also shown. The N297-linked N-glycan structure attached to the IgG1 Fc is shown in red and the β2-microglobulin subunit of FcRn is shown in yellow. The figure was made in PyMOL using crystallographic data from PDB entry 4NOU[21]. **b** Illustration showing the human FcRn binding ELISA setup. **c**, **d** ELISA showing binding of NIP IgG1 WT, REW, and H310A to human FcRn at pH 6.0 and 7.4, shown as mean±s.d of duplicates. **e** Illustration showing the human FcRn SPR kinetics

assay. **f**, **g** SPR sensorgrams showing binding of monomeric human FcRn to immobilized anti-NIP IgG1 WT or REW at pH 6.0. **h** Illustration outlining the HERA cellular assay. HERA showing (**i**) uptake, (**j**) rescue from intracellular degradation (recycling), (**k**) residual amounts and (**l**) the derived HERA scores for the antibodies, shown as mean±s.d of triplicates from two independent experiments. **m** Illustration outlining the in vivo plasma half-life experiments. Plasma clearance of (**n**) anti-NIP and (**o**) anti-SARS-CoV-2 mAb4 IgG1-WT and REW (5 mg/kg) in mice pre-loaded with 500 mg/kg IVIg, shown as mean±s.d of percent antibody remaining in plasma over time ($n = 5$ animals per group). **i**–**l** Unpaired two-sided $t$-test, (**n**, **o**) RM Two-way ANOVA with Šídák's multiple comparison test. Source data are provided as a Source Data file. **b**, **e**, **h**, **m** were created using BioRender.com.

supporting that increased amount of REW in lungs is a result of higher plasma concentration rather than specific FcRn mediated transport in this organ. This agrees with previous studies in lungs for a half-life extended IgG1 variant in non-human primates[24,37]. As FcRn has also been implicated in active transport of IgG Fc based biologics from the airway lumen to the systemic circulation[38], we next administered anti-NIP IgG1 WT and REW intranasally (i.n.) and measured their concentrations in plasma 24 h post-delivery (Fig. 2d). This demonstrated that 1.6-fold more of IgG1-REW reached the circulatory system (Fig. 2e). In line with this, IgG1-REW was transcytosed more efficiently across polarized monolayers of the human colon-derived epithelial cell line T84, expressing endogenous FcRn, as well as across a Madine Darby Canine Kidney (MDCK) cell line overexpressing the receptor in a transwell system (Fig. 2f), where apically directed transport was more efficient than in the opposite direction in both cell lines (Fig. 2g, h). These results motivated us to explore REW as part of a subunit vaccine by fusing the globular domain of hemagglutinin from *influenza A* (HA; H1N1 A/Puerto Rico/8/1934 (PR8)) to WT and REW IgG1 Fc (Fig. S3p). The monovalent fusions were produced, bound FcRn pH-dependently (Fig. S3q–s) and were administered i.n. to human FcRn-expressing mice together with CpG adjuvant, followed by a 10% booster dose after 3 weeks. Three weeks after the boost, the mice were challenged i.n. with a 5x lethal dose of H1N1 PR8 virus (Fig. 2i). While 60% of the mice vaccinated with the WT HA-Fc fusion succumbed within 8 days, all mice vaccinated with REW-containing HA-Fc were fully protected

(Fig. 2j, Fig. S3t), and showed a modest increase in HA-specific antibody titer, although not statistically significant (Fig. S3u).

We then investigated maternal to fetal transport in an ex vivo human placental perfusion model[39]. To do so, WT and REW anti-NIP IgG1 variants were mixed 1:1 with infliximab (anti-TNF alpha IgG1) and then added to the maternal reservoir before samples were collected from both the maternal and fetal side over time (Fig. 2k). Quantification of the antibodies transported demonstrated that REW reached the fetal side two times more efficiently at endpoint (Fig. 2l). Further, transport of infliximab was reduced in the presence of IgG1-REW, supporting a competitive advantage for the Fc-engineered antibody (Fig. S3v).

## REW enhances complement activation in an on-target manner

Our structure-based approach did not only aim to improve engagement of human FcRn, but also the capacity to form Fc:Fc interactions in an on-target manner to facilitate hexamer Fc formation and complement activation[40] (Fig. 3a, SI Text 3, Fig. S4). To test this, we measured binding of complement factors following capture of equal titrated amounts of the NIP IgG1 antibodies on NIP-conjugated bovine serum albumin (BSA) coated in ELISA wells, which revealed considerably enhanced C1q binding (Fig. 3b, c), deposition of the complement components C3, C4 and C5 (Fig. S5a–d) as well as formation of the terminal complement complex (TCC) (C5bC9) (Fig. 3d, e). No activity was measured for an Fc-engineered effector-negative IgG1

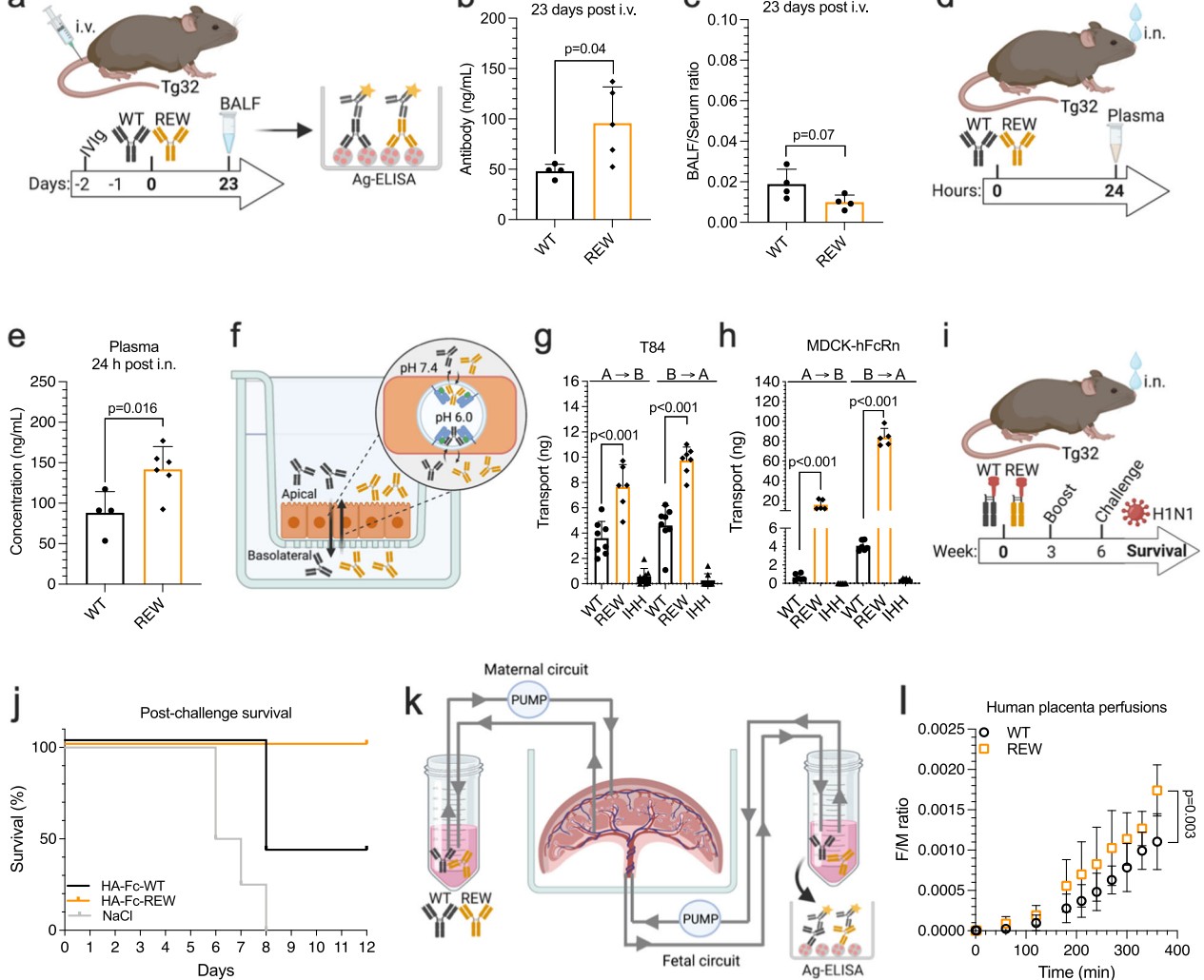

**Fig. 2 | REW improves bioavailability, transmucosal delivery and vaccination.**
**a** Illustration outlining the in vivo lung localization experiment in Tg32 mice.
**b** Amount of NIP IgG1 WT and REW in BALF 23 days post i.v. administration of 5 mg/kg of the antibody variants in IVIg (500 mg/kg) pre-loaded mice. Shown as mean ± s.e.m ($n = 4$ animals (WT) and $n = 5$ animals (REW)). **c** BALF/plasma ratio of NIP IgG1 WT and REW, 23 days post i.v. administration. Shown as mean ± s.d ($n = 4$ animals per group). **d** Illustration outlining i.n. delivery experiment in Tg32 mice.
**e** Plasma concentration of NIP IgG1 WT and REW 24 h post i.n. delivery (2.23 mg/kg) in Tg32 mice. Shown as mean ± s.d. ($n = 4$ animals (WT) or $n = 6$ animals (REW)).
**f** Illustration outlining FcRn mediated transcytosis experiments in a transwell system. **g** Apical to basolateral (A→B) and basolateral to apical (B→A) directed transport of NIP IgG1 WT, REW and IHH across polarized human T84 cells, shown as mean ± s.d ($n = 8$ independent monolayers from 2 independent experiments) (two data points was excluded from REW A→B and one data point excluded for REW

(B→A) due to disrupted cell monolayers). **h** Apical to basolateral (A→B) and basolateral to apical (B→A) directed transport of NIP IgG1 WT, REW and IHH across MDCK-hFcRn cells, show as mean ± s.d ($n = 6$ independent cellular monolayers from 2 independent experiments) (one data point were excluded from REW A→B and REW (B→A) due to disrupted cell monolayers). **i** Illustration outlining mucosal vaccine and challenge experiment in Tg32 mice. **j** Percent survival of Tg32 mice i.n. vaccinated with HA WT and REW monovalent Fc fusions or NaCl control following challenge with a lethal dose of H1N1 virus. **k** Illustration showing the ex vivo human placental perfusion model used to measure maternal-to-fetal transport of anti-NIP IgG1 WT and REW. **l** Ex vivo human placental perfusion model showing maternal/fetal (FM) transport ratio of NIP IgG1 WT and REW, shown as mean ± s.d. ($n = 4$ placentas per group). **b, c, e, g, h** Unpaired two-tailed *t*-test, (**l**) Two-tailed Wilcoxon *t*-test. Source data are provided as a Source Data file. (**a, d, f, l, k**) were created using BioRender.com.

(L234A/L235A/P329G) (PGLALA)[41]. The REW variant also showed enhanced binding to mouse C1q (Fig. S5e, f).

As a matter of safety, antibody mediated complement activity should only occur in an on-target manner and not spontaneously in solution. To ensure that this was indeed the case, WT, REW and PGLALA anti-NIP IgG1 variants were coated directly in ELISA wells in parallel with an IgG1 variant containing the E345R/E430G/S440Y (RGY) amino acid substitutions, which is known to form solution phase hexamers[40]. In line with this, only IgG1-RGY efficiently bound C1q in normal human serum (NHS) when coated randomly to a surface (Fig. 3f). In addition, to confirm that REW did not induce formation of higher order assemblies followed by complement activation in the

absence of cognate antigen, the antibodies were incubated in NHS at 37 °C at a concentration of 100 μg/mL before formation of IgG complexes and C4d was measured in ELISA (Fig. 3g), which showed that only RGY formed IgG complexes and triggered complement activation in solution (Fig. 3h, i).

To address potential off-target complement activation in vivo, NIP specific IgG1 WT, REW and RGY were administered to Tg32 mice before collection of plasma after 24 h (Fig. S5g) Quantification of the antibody levels (Fig. S5h) and subsequent analysis of human IgG normalized plasma samples using a mouse C1q specific ELISA (Fig. S5i) showed that only RGY bound mouse C1q in vivo while no difference was measured between WT, REW and plasma from non-treated animals

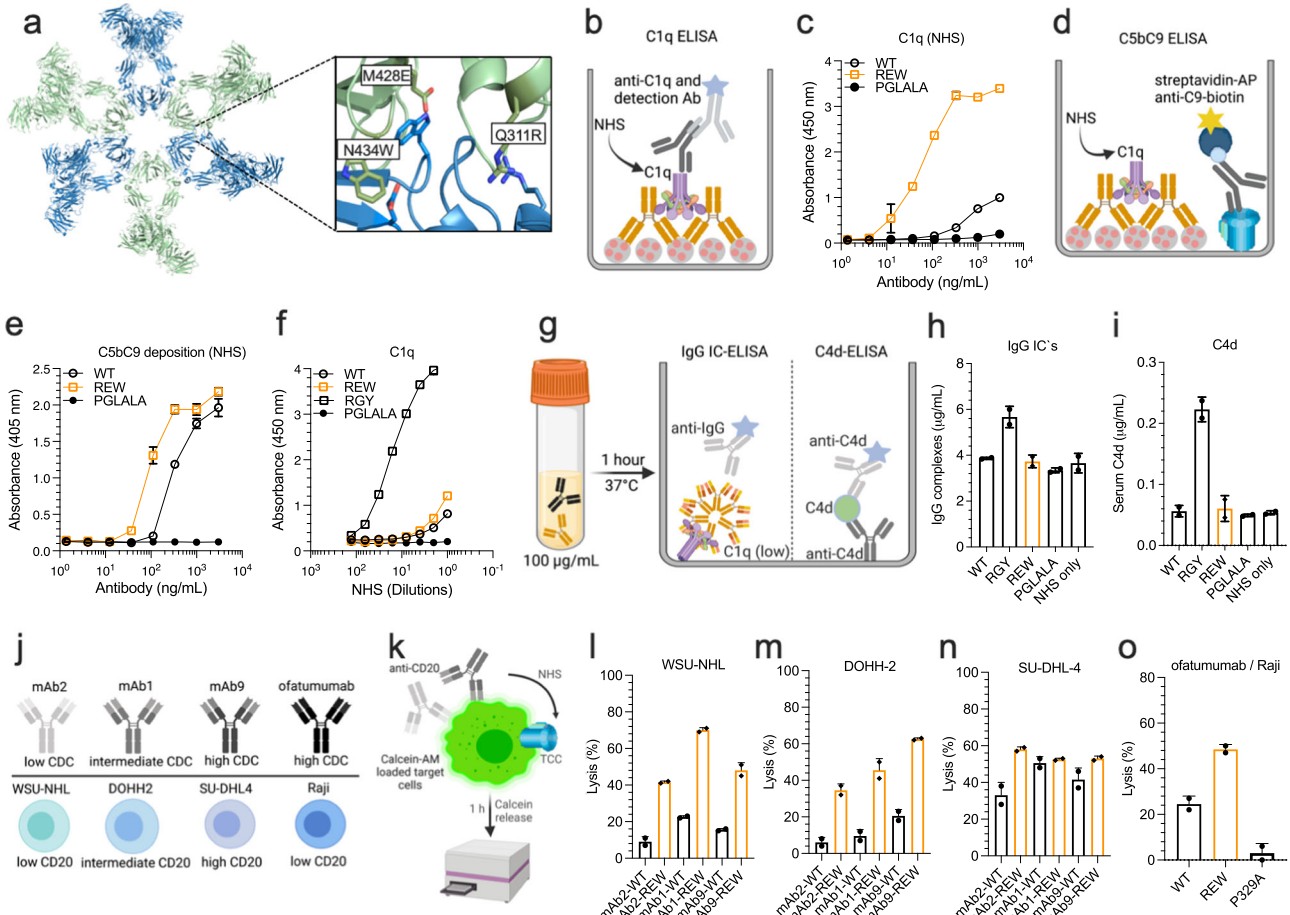

**Fig. 3 | REW potentiates on-target complement activation and killing of cancer cells. a** Structural model of hexameric IgG with a close-up of the Fc:Fc interface. The REW amino acid substitutions are shown. The model was made in PyMOL using crystallographic data from PDB entry 1HZH[40,81]. **b** Illustration showing the human C1q binding ELISA setup. **c** ELISA binding of C1q from NHS to titrated amounts of antigen captured anti-NIP IgG1 WT, REW and PGLALA, shown as mean ± s.d of duplicates. **d** Illustration showing the C5bC9 deposition ELISA setup. **e** ELISA showing C5bC9 (TCC) formation from NHS as a function of titrated amounts of antigen captured anti-NIP IgG1 WT, REW and PGLALA, shown as mean ±s.d of duplicates. **f** ELISA binding of C1q from NHS to titrated amounts of randomly immobilized anti-NIP IgG1 WT, REW, RGY and PGLALA, shown as mean ± s.d of duplicates. **g** Illustration outlining solution phase complement activation assays.

**h** Amount of IgG complexes and (**i**) C4d in NHS incubated with 100 µg/mL of anti-NIP WT, REW, RGY and PGLALA or NHS only at 37 °C for 1 h, shown as mean ± s.d of duplicates. **j** Overview of anti-CD20 antibodies and CD20+ cell lines used in CDC assays and (**k**) Calcein-AM release assay used to measure antibody dependent CDC of cancer cells. **l−n** CDC activity of anti-CD20 mAb2 (low CDC), mAb1 (intermediate CDC), mAb9 (high CDC) WT and REW against WSU-NHL (low CD20), DOHH-2 (intermediate CD20) and SU-DHL-4 (high CD20) lymphoma target cells, shown as mean±s.d of duplicates. **o** CDC activity of recombinant forms of ofatumumab (anti-CD20, IgG1) WT, REW and P329A against Raji target cells, shown as mean ± s.d of duplicates. Source data are provided as a Source Data file. **a, c, g** were created using BioRender.com.

(Fig. S3j). In line with this, only RGY led to elevated levels of mouse C3a (Fig. S5k, l).

## REW enhances complement mediated killing of cancer cell lines
Next, we investigated the ability of IgG1 REW to induce CDC of CD20-expressing lymphoma cell lines. To do so, we first made recombinant variants (WT and REW) of three recently developed anti-CD20 IgG1 antibodies, with distinct ability to mediate CDC, namely mAb2 (low CDC), mAb1 (intermediate CDC) and mAb9 (high CDC)[42]. Using a Calcein-AM release assay, the antibodies were screened for their ability to mediate CDC against three lymphoma cell lines; WSU-NHL (low CD20), DOHH2 (intermediate CD20) and SU-DHL-4 (high CD20), reported to express similar levels of agonistic complement receptors[43] (Fig. 3j). The REW containing variants showed considerably enhanced CDC activity, with the most prominent effects measured at low and intermediate CD20 levels, while the effect of REW waned at higher CD20 levels and increased inherent CDC activity of the antibody variants (Fig. 3l−n). The antibody variants with the C1q non-binding substitutions PGLALA showed no or low CDC activity (Fig. S5m). In line with this, ELISA results revealed that anti-NIP IgG1 REW engages C1q

more efficiently than WT when captured on different cognate antigen densities (Fig. S5n−q). Finally, we addressed the ability of IgG1 REW to induce CDC of "hard-to-kill" CD20+ Raji cells using recombinant variants of the clinically approved anti-CD20 IgG1 ofatumumab (WT, REW and the effector-negative P329A). Again, the results demonstrated 2-fold potentiated CDC activity of the REW-containing variant compared to WT (Fig. 3o).

## REW potently induces phagocytosis of gram positive bacteria and killing of gram negative bacteria
The Fc binding site for FcRn overlaps with the surface area required for Fc:Fc mediated hexamer formation[40], which is also a hot spot binding site for bacterial defense proteins, such as protein A (SpA) from gram-positive *Staphylococcus aureus* (*S. aureus*)[44]. Recently, binding of SpA to IgG1 Fc was shown to prevent hexamer formation, complement activation and subsequent phagocytic killing by polymorphonuclear leukocytes (PMNs)[45]. Importantly, we here demonstrate that the REW substitutions strongly attenuate binding to SpA (Fig S5r, s), and

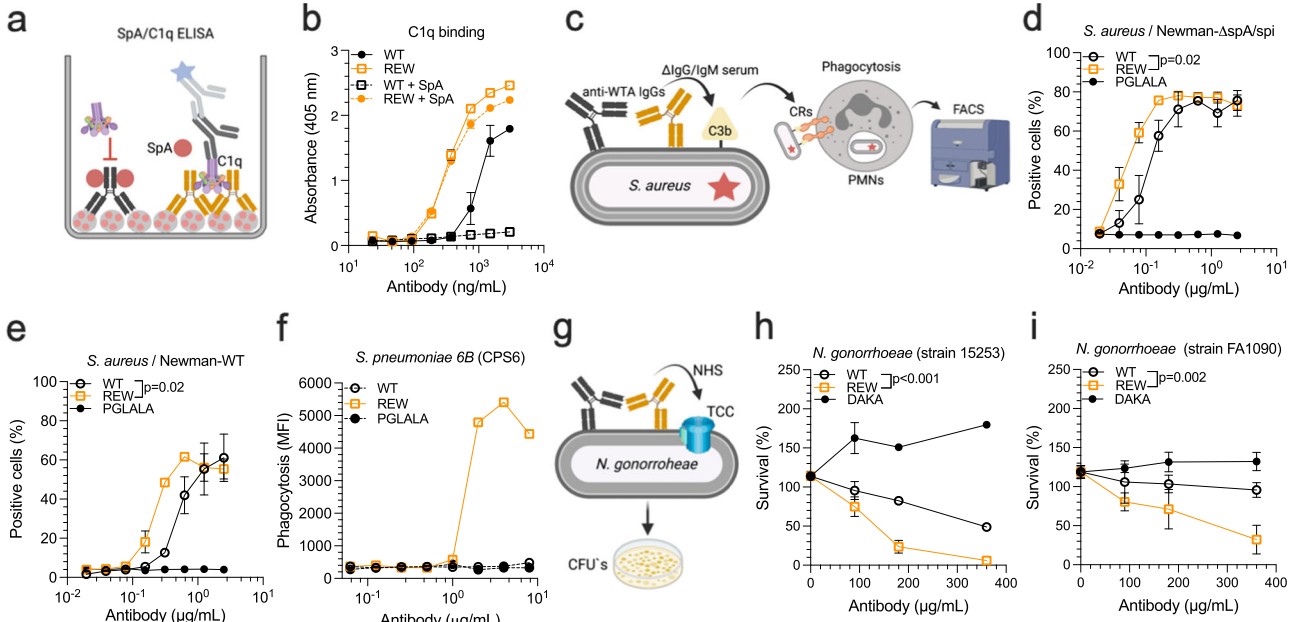

**Fig. 4 | REW enhances phagocytosis of gram-positive and killing of gram-negative bacteria. a** Illustration showing the ELISA setup used to measure C1q binding to anti-NIP IgG1 WT and REW in presence and absence of a molar excess of SpA. **b** ELISA showing binding of C1q to antigen captured anti-NIP IgG1 WT and REW in presence and absence of 5-fold molar excess of SpA, shown as mean±s.d of duplicates. **c** Illustration outlining *S. aureus* PMN phagocytosis experiments of anti-WTA IgG1 variants in presence of ΔIgG/IgM human serum. **d** PMN phagocytosis of *S. aureus* strain Newman-ΔspA/spi and (**e**) Newman-WT bound by titrated amounts of anti-WTA (clone 4497) IgG1 WT, REW or PGLALA in presence of IgG/IgM depleted human serum, shown as mean±s.d of triplicates. **f** PMN phagocytosis of *S. pneumoniae* serotype 6B (CSP6) in presence of ΔIgG/IgM human serum, shown as mean ± s.d of duplicates. **g** Illustration showing the *N. gonorrhoeae* CDC assay. **h**, **i** CDC activity against *N. gonorrhoeae* strain 15253 and strain FA1090 bound by titrated amounts of humanized anti-lipo-oligosaccharide IgG1 WT, REW or DAKA (Humab 2C7), shown as mean ± s.d of triplicates. **d**, **e**, **h**, **i** RM Two-way ANOVA with Šídák's multiple comparison test. Source data are provided as a Source Data file. **b**, **d**, **g**, **j**, **k** were created using BioRender.com.

accordingly, IgG1-REW could efficiently engage C1q, even in the presence of 5-molar excess of SpA, which prevented binding to WT IgG1 (Fig. 4a, b). We then addressed whether REW could enhance antibody-mediated bacterial phagocytosis. By taking advantage of an IgG1 antibody (clone 4497) specific for wall teichoic acid (WTA) on *S. aureus*[45,46], we first showed that the WT, REW and PGLALA-containing versions bound equally well to the bacterial surface of an *S. aureus* strain lacking the two IgG binding molecules SpA and Spi (NewmanΔspA/spi) (Fig. S5t, u). Next, we measured antibody-dependent complement-mediated phagocytosis of NewmanΔspA/spi. The bacteria were incubated with human PMNs in the presence of IgM/IgG-depleted human serum followed by measurement of phagocytosis (Fig. 4c). The results revealed increased phagocytotic activity of IgG1-REW opsonized bacteria, with a more than 10-fold lower EC50 than for the WT antibody (Fig. 4d, Table S5). When the experiment was repeated using *S. aureus* expressing SpA on its surface (Newman-WT), REW maintained its superior activity with a 5-fold lower EC50 value than WT (Fig. 4e, Table S6). We further tested the ability of REW to enhance phagocytosis of *Streptococcus pneumoniae* (*S. pneumoniae*) when introduced into a capsule polysaccharide specific IgG1 (Dob1) with poor inherent ability to activate complement[47,48]. Again, REW showed potent ability to mediate complement-dependent phagocytosis by PMNs of capsular serotype 6B (CSP6) *S. pneumoniae* (Fig. 4f). No activity was measured for WT and IgG1-PGLALA despite that they bound equally well to the bacterial surface (Fig. S5v).

Additionally, we explored REW in the context of the gram-negative *Neisseria gonorrhoeae* (*N. gonorrhoeae*), in which antibody-mediated killing has been shown to occur independently of PMNs and instead be driven by direct TCC mediated lysis[49]. To do so, we used a humanized version of mAb 2C7 specific for the gonococcal lipo-oligosaccharide (LOS) containing IgG1 Fc (Humab 2C7) that has been shown to induce complement-mediated killing of *N. gonorrhoeae* in

mice[49]. While the REW, D270A/K322A (DAKA; complement null) and WT Humab 2C7 versions bound equally well to the bacterial surface of *N. gonorrhoeae* (strain 15253) (Fig. S5w), the binding of REW resulted in enhanced killing, as bacterial growth was reduced by 40-60% compared to WT at the highest antibody concentrations (Fig. 4g, h). *N. gonorrhoeae* strain FA1090, which is more resistant to Humab 2C7 mediated CDC due to differences in LOS glycan composition[49], was efficiently killed by 2C7-REW while the WT showed reduced activity (Fig. 4i).

### REW combined with afucosylation enhances ADCC
To address the effect of the REW substitutions on FcγR engagement, we first used anti-CD20 mAb2 IgG1 to measure macrophage-mediated ADCP against Raji target cells. In this case, the REW variant performed equally well as that of WT (Fig. S6a, b). To measure ADCC, we first made a REW version engineered for low content of N297-linked fucose which considerably increased binding to FcγRIIIa (Fig. 5a–c). Next ADCC activity against cell lines expressing low (WSU-NHL and Carnaval) or high (SU-DHL-4) levels of CD20 was measured in presence of anti-CD20 mAb2 IgG1 variants using human mononuclear cells (MNCs) in a [51]Cr release assay[50] (Fig. 5d, e). Again, equal activity was measured for WT and REW, however, when the REW version was engineered for low content of N297-linked fucose, substantially increased ADCC activity was observed (Fig. 5f–h). Importantly, afucosylation of IgG1-REW did not compromise its long plasma half-life (16.3 days) compared to WT (9.2 days) in human FcRn-expressing mice (Fig. 5i, j).

### REW-IgG subclasses show favorable FcRn engagement but distinct effector functions
As the REW substitutions are conserved among the four human IgG subclasses, we made REW engineered anti-NIP versions, in addition to a recombinant form of the IgG1 Fc fusion molecule etanercept (Fig. 5k).

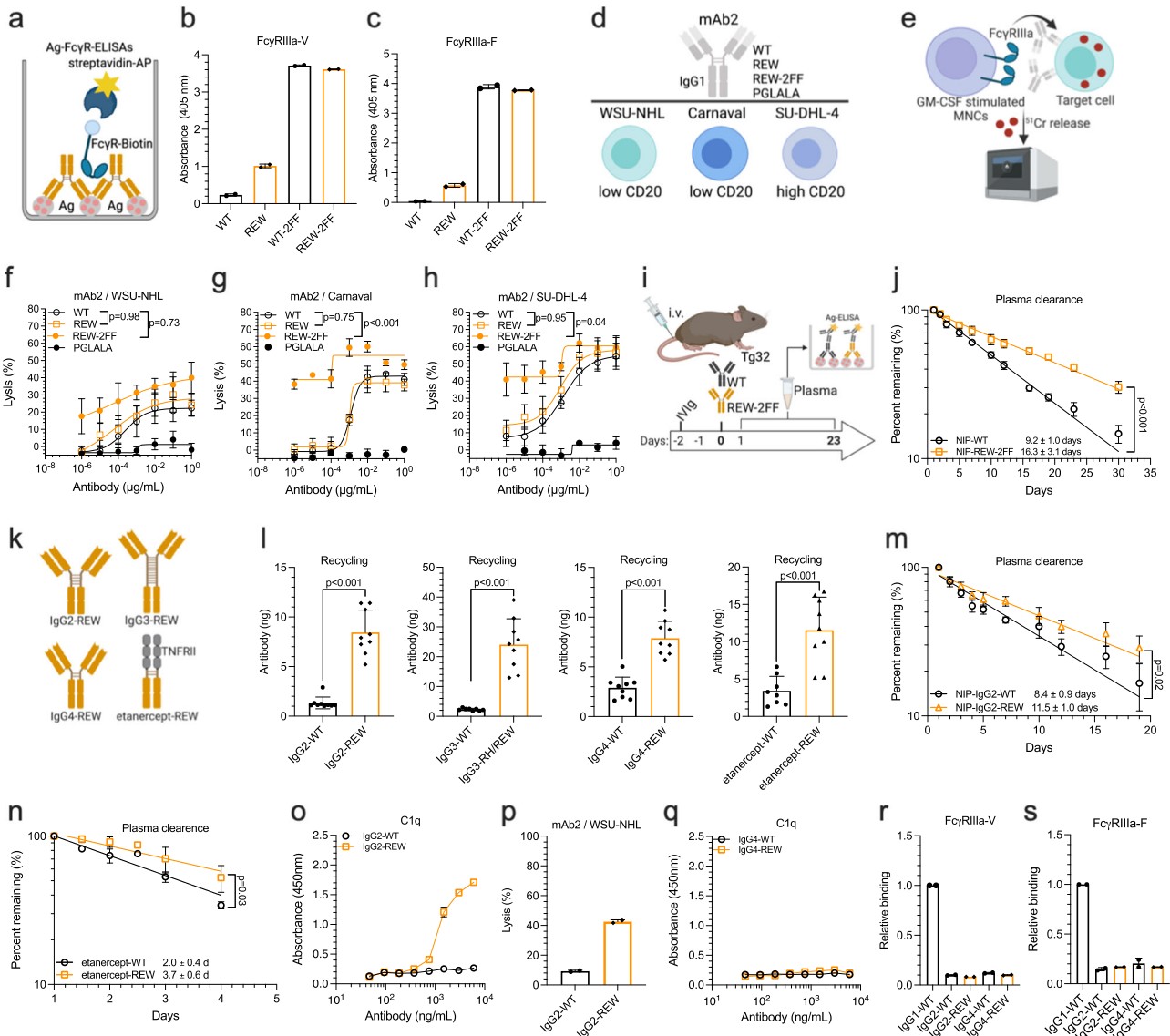

**Fig. 5 | Enhanced ADCC by *N*-glycan-engineering without compromising plasma half-life and IgG subclass specific effect of REW on C1q binding.**
**a** Illustration of ELISA setup and binding of anti-NIP IgG1 WT, REW and low-fucose (2FF) WT and REW to (**b**) FcγRIIIa-V158 and (**c**) FcγRIIIa-F158, shown as mean ± s.d of duplicates. **d** Overview of anti-CD20 IgG1 mAb2 variants and B cell lines used to measure ADCC activity. **e** The $^{51}$Cr release assay used to measure ADCC activity of MNCs against target cell lines in presence of the antibodies. **f**–**h** MNC mediated ADCC of WSU-NHL (low CD20), Carnaval (low CD20) and SU-DHL-4 (high CD20) lymphoma cell lines by the WT mAb2 compared with the Fc-engineered versions; REW, REW-2FF and PGLALA, shown as mean±s.e.m. from 5 replicates performed in parallel. **i** Experimental outline and (**j**) plasma clearance of anti-NIP IgG1 WT and REW-2FF in IVIg pre-loaded (500 mg/kg) Tg32 mice, shown as mean±s.d of percent antibody remaining in plasma over time (*n* = 5 animals per group). **k** Illustration showing REW-engineered IgG subclasses and a recombinant form of Fc-fusion etanercept. **l** Cellular FcRn mediated rescue from intracellular degradation of WT and REW engineered IgG subclasses and Fc-fusion etanercept in HERA, shown as mean±s.d of triplicates from three independent experiments. **m** Plasma clearance of anti-NIP IgG2 WT and REW following i.v. administration (5 mg/kg) in IVIg pre-loaded (500 mg/kg) Tg32 mice, shown as mean ± s.d of percent antibody remaining in plasma over time (*n* = 4 animals per group). **n** Plasma clearance of recombinant etanercept WT and REW following i.v. administration (5 mg/kg) in Tg32 mice, shown as mean ± s.d of percent antibody remaining in plasma over time (*n* = 5 animals per group). **o** ELISA showing binding of human C1q to titrated amounts of antigen captured anti-NIP IgG2 WT and REW, shown as mean±s.d of duplicates. **p** CDC activity of mAb2 anti-CD20 IgG2 WT and REW against WSU-NHL (low CD20) using the $^{51}$Cr release assay, shown as mean±s.d of duplicates. **q** ELISA showing binding of human C1q to titrated amounts of antigen captured anti-NIP IgG4 WT and REW, shown as mean±s.d of duplicates. **r**, **s** ELISAs showing relative binding of FcγRIIIa-V158 or FcγRIIIa-F158 to antigen captured anti-NIP IgG2 and IgG4 WT and REW, shown as relative binding compared to anti-NIP IgG1-WT, mean ± s.d of duplicates. **f**, **g**, **h**, **j**, **m**, **n** RM Two-way ANOVA with Šídák's multiple comparison test. **l** Unpaired two-tailed *t*-test. Source data are provided as a Source Data file. **a**, **d**, **e**, **l**, **k** were created using BioRender.com.

In all cases, REW improved pH-dependent binding to human FcRn (Fig. S7, Table S7). This resulted in more efficient rescue from intracellular degradation in HERA (Fig. 5l) and extended plasma half-life for IgG2-REW (11.5 days) over IgG2-WT (8.4 days) and the engineered Fc-fusion (3.7 days) over the non-engineered version (2.0 days) (Fig. 5m, n). We then addressed the ability of the engineered IgG subclasses to engage C1q, which revealed that IgG2 turned into a complement activator that could mediate CDC against CD20 low WSU-NHL cells (Fig. 5o, p). In contrast, IgG4-REW remained inert as it did not engage C1q (Fig. 5q). In addition, IgG2-REW and IgG4-REW did not bind FcγRIIIa (Fig. 5r, s).

Finally, to further assess immunogenicity, NetMHC4.1 was used to predict T-cell epitopes and IEDB analysis was used to predict antibody epitopes, in which the REW amino acid substitutions did not

significantly increase the number of potentially immunogenic peptides or epitopes compared with WT IgG1 and other Fc-engineering strategies for half-life modulation (Tables S11–S12). REW-containing IgG1 also showed reduced reactivity to rheumatoid factor antibodies (SI Text 4, Fig. S8), known to negatively affect the plasma half-life of Fc-engineered antibody variants[51].

## Discussion

Therapeutic or prophylactic monoclonal antibodies are engineered on a case-to-case basis and require careful consideration regarding which effector functions are to be maintained, enhanced, or abolished. This is to provide potent and specific treatment with minimal risk of side effects[3,52]. In all cases, the pharmacokinetic profile will guide dosing and frequency of administration, with the aim to secure high concentration of active antibody in blood and tissue, with as much as possible reaching its target over time.

While several IgG Fc variants with increased plasma half-life have been developed, and some have successfully reached the clinic[23–25,36,53,54], few have been Fc-engineered by a small number of amino acid substitutions that increase plasma half-life, improve biodistribution and specific effector functions, such as complement activity. Here, we report on such a dual engineering approach, where the REW substitutions in IgG1 resulted in favorable pharmacokinetics in human FcRn expressing mice, combined with increased ability to activate the classical complement pathway in an on-target manner. The design was based on the fact that FcRn binds at the $C_H2$-$C_H3$ elbow region at a site that overlaps with that of Fc:Fc interactions required for formation of on-target hexamers, pivotal for complement activation[40]. When benchmarked against other IgG1 Fc variants engineered for altered FcRn binding and plasma half-life in vitro and in vivo, REW performed better or on par with these (SI Text 5, Fig. S9a–n, Tables S8–S9). Comparing REW with an engineered Fc variant with enhanced CDC activity, but without half-life extension[43,55], equal CDC activity against a "hard-to-kill" cell line was measured for recombinant versions of anti-CD20 ofatumumab (SI Text 5, Fig. S9o–p, Table S10). The REW variant also showed enhanced C1q binding compared to other half-life extended IgG1 Fc-engineered variants (SI Text 5, Fig. S9q). This was also the case for the low affinity human FcγRs when captured on cognate antigen in ELISA but not in the absence of antigen in SPR (SI Text 5, Fig. S10).

Another unique property of REW is that human FcRn binding and cellular transport properties are enhanced for the four human IgG subclasses while their ability to engage C1q and FcγRIIIa occurs in a subclass specific manner. For instance, while REW-containing IgG2 is equipped with the capacity to mediate induction of classical complement activation, IgG4 REW remains inert in this regard, which is a differentiator compared to other reported Fc-engineered strategies. REW was also shown to prolong the plasma half-life of a recombinant version of the Fc-fusion etanercept. In addition, REW enhanced the transport across cell layers and mucosal barriers upon intranasal delivery in mice, which was also the case for an influenza A HA-fused IgG1 Fc-fusion that was shown to protect against a deadly dose of the virus in the presence of adjuvant. While a modest increase in HA specific mouse antibodies was measured in the mice vaccinated with HA-REW compared with those given HA-WT, the difference was not statistically significant. However, the levels detected were low, which is explained by the fact that human FcRn binds poorly to mouse IgG antibodies, resulting in rapid intracellular degradation[31,56] of the generated antibodies. Thus, the mouse model used is suboptimal for studies of human Fc-engineered vaccine formats. Although this preliminary finding is promising, the vaccine concept should be explored in a more suitable model like in mice that are transgenic for both human FcRn and human IgG[25,57]. Hence, REW is a versatile platform technology that can be used to select the most favorable subclass or Fc fusion design depending on target and indication.

Exploration of the REW toolbox should be attractive in design of antibodies to fight cancer and to improving management of chronic diseases. However, to test the concepts in vivo, and in particular, in the case of cancer, the mice should not only express human FcRn but also all the human forms of FcγRs to fully address the therapeutic effect of enhanced half-life and biodistribution combined with tailored engagement of effector molecules and immune cells. The REW technology may also be suited for antibodies tailored for targeting infectious diseases. The reason for this is that the duration of action of a given antibody, or cocktail of antibodies, both systemically and at mucosal sites, is key to secure effective long-acting treatment[11]. Antibodies engineered to be both very potent and long-acting could likely be given in smaller doses and at longer intervals. Hopefully, these factors could make the antibodies more affordable for global access, including use in low and middle-income countries[58]. For instance, with the lack of an effective HIV vaccine or weak responses to SARS-CoV-2 vaccines, broadly neutralizing and long-acting anti-HIV and anti-SARS-CoV-2 antibody cocktails may provide a way to prevent or control disease[59,60].

Related to this is the global concern for AMR bacteria[61]. Resistance is associated with respiratory infections of for instance *S. pneumoniae and S. aureus*. While antibody treatment to combat viral infection has been boosted because of the SARS-CoV-2 pandemic, development in context of pathogenic bacteria is underexplored[6,7]. Importantly, bacteria have developed strategies to circumvent attack by antibodies. Here, we show that REW reduces the ability of bacterial defense protein SpA from *S. aureus* to bind to the Fc at a site that overlaps with that of FcRn and the Fc:Fc contacts needed for efficient complement activation[45]. Thus, when we introduced REW into IgG1 antibodies targeting this bacteria, increased complement dependent phagocytosis was observed. In addition, we show that a REW-containing *S. pneumonia* specific IgG1 antibody could overcome capsular resistance to complement dependent phagocytosis[62]. Notably, the REW substitutions also reduce binding to *S. pneumoniae* defense protein G (Fig. S11). Enhanced killing was also the case when REW was tested for antibody-mediated CDC against gram-negative *N. gonorrhoeae*. These results strongly motivate further development of REW-containing antibodies against AMR bacteria, which should involve establishment of infection mouse models expressing human FcRn.

## Methods

### Cell culture

Human embryonic kidney (HEK) 293E (ATCC, CRL-1573), Raji cells (ATCC, CCL-86), WSU-NHL (DSMZ GmbH, ACC 58), DOHH-2 (DSMZ GmbH, ACC 47) and SU-DHL-4 (DSMZ GmbH, ACC 495) were maintained in RPMI 1640 medium with L-glutamine (ThermoFisher) all supplemented with 10% heat-inactivated (HI) fetal bovine serum (FCS) (Merck), 25 µg/mL streptomycin and 25 U/mL penicillin (ThermoFisher). T84 cells (ATCC, CCL-248) were maintained in HAM's F12/DMEM (1:1) with L-glutamine (ThermoFisher) supplemented with 10% HI-FCS, 25 µg/mL streptomycin and 25 U/mL penicillin (ThermoFisher). HMEC1-hFcRn cells[63] were maintained in MCDB 131 medium (ThermoFisher) supplemented with 10% HI-FCS, 2 mM L-glutamine, 25 µg/mL streptomycin, 25 U/mL penicillin, 10 ng/mL mouse epidermal growth factor (mEGFR) (Peprotech) and 1 µg/mL hydrocortisone. MDCK-hFcRn cells were generated Drs. Jens Fisher and Alex Haas (Roche Pharma Research and Early Development) and cultured in DMEM supplemented with 10% HI-FCS, 25 µg/mL streptomycin, 25 U/mL penicillin, and 300 µg/mL G418 (Sigma-Aldrich). Expi293 cells (ThermoFisher, A14527) were maintained in Expi293 medium (ThermoFisher) supplemented with 10% HI-FCS, 25 µg/mL streptomycin and 25 U/mL penicillin (ThermoFisher). All cell lines were kept in a humidified 37 °C/5% $CO_2$ incubator, except Expi293 cells that were kept in a humidified 37 °C/8% $CO_2$ incubator on an orbital shaker (125 rpm).

## Antibodies

Expression vectors encoding the heavy chain (HC) and light chain (LC) of NIP[64], anti-CD20 (mAb2, mAb1, mAb9 and ofatumumab)[42,65], anti-*S. aureus* WTA (4497)[46], anti-*S. pneumonia* (Dob1)[48] and anti-*N. gonorrhoeae* (2C7)[66] specific mouse-human chimeric or human WT IgG variants were generated by synthesizing cDNA followed by subcloning into the described pLNOH2/pLNOk vector system (Genscript Inc). SARS-CoV-2 specific mAb4 (THSC20.HVTR04)[33] WT IgG1 were generated by synthesizing the V-region followed by subcloning into the pFUSE expression vector system (Invivogen) (Genscript Inc). Vectors encoding IgG variants with site-specific substitutions were generated by site-directed mutagenesis (Genscript Inc). All IgG variants or recombinant Fc fragments were produced in HEK293E cells or Expi-CHO (Humab 2C7) by transient transfection using Lipofectamine© 2000 (ThermoFisher). The IgG variants were purified using CaptureSelect™ C$_H$1 columns (Thermo Fisher) prior to size exclusion chromatography (SEC) to isolate monomeric fractions using a Superdex 200 Increase 10/300 column (Cytiva Life Sciences) with an ÄKTA Avant 25 (Cytiva Life Sciences). Purified IgG variants were concentrated using Amicon Ultra 50 K spin columns (Merck) and stored in either 1x phosphate buffered saline (PBS) (NIP, anti-CD20, anti-*S. aureus* WTA (4497), anti-*S. pneumoniae 6B* (CPS6) and anti-*N. gonorrhoeae* (2C7) or 20 mM TRIS-HCl, 140 mM NaCl, pH 5.6 (mAb4). Protein concentrations were determined using a DS-11 spectrophotometer (DeNovix). For production yield experiments, NIP IgG1 WT and REW were produced in Expi293 cells using the Expifectamine© Transfection Kit (Thermo Fisher) and purified as above.

## Recombinant FcRn

Truncated soluble and Glutathione S-transferase (GST) tagged human FcRn lacking the transmembrane domain (hFcRn-GST) was produced in HEK293E cells and purified on a GSTrap column (Cytiva Life Sciences)[67]. Truncated soluble and biotinylated human and mouse FcRn forms lacking the transmembrane domain were acquired from Immunitrack Inc. Truncated soluble His$_{6x}$ tagged human FcRn lacking the transmembrane domain (hFcRn-His) was produced in a Baculovirus expression system and purified using a HisTrap HP column (Cytiva Life Sciences)[68,69]. The Baculovirus stock was a kind gift from Dr. Sally Ward (University of Southampton). Monomeric fractions of hFcRn-His were isolated by SEC using a Superdex 200 Increase 10/300 column (Cytiva Life Sciences) with an ÄKTA Avant 25 (Cytiva Life Sciences).

## FcRn binding ELISAs

Stated titration series of NIP specific IgG variants were captured on bovine serum albumin (BSA) conjugated to NIP (1:25 ratio) (BSA-NIP(25)) coated at 1 µg/mL/100 µl in 96-well EIA/RIA plates (Corning) and blocked with 4% skimmed milk (S) in PBS overnight (ON). All remaining steps were performed using either phosphate buffer pH 6.0 with 4% S and 0.05% Tween 20 (T) or PBS/T/S pH 7.4 as dilution and wash buffers. Then FcRn-GST (0.25 µg/ml or 2.0 µg/mL), biotinylated human or mouse FcRn (Immunitrack) (2.5 µg/mL or 0.25 µg/mL) were added and incubated for 1 h at RT. When 0.25 µg/mL biotinylated FcRn were used at pH 7.4, the receptor was pre-incubated with alkaline phosphatase (AP) conjugated streptavidin (1:5000 in PBS/T/S) to increase sensitivity. Bound FcRn was detected either by a horseradish peroxidase (HRP)-conjugated anti-GST antibody from goat (Rockland Immunochemicals, 200-301-200) (diluted 1:5000 in PBS/T/S) or streptavidin-AP (diluted 1:5000 in PBS/T/S) and visualized by addition of TMB substrate (CalBiochem) or p-nitrophenylphosphate (AP substrate) (Sigma-Aldrich) diluted to 10 µg/mL in diethanolamine buffer. The HRP reaction was terminated by addition of 1 M HCl and the 450 nM (HRP) or 405 nm (AP) absorption values was recorded using a Sunrise Spectrophotometer (TECAN).

## FcRn SPR binding assay

A Biacore 3000 (Cytiva Life Sciences) was used to couple NIP IgG1 variants (300 resonance units (RU) to CM5 sensor chips using an amine coupling kit (Cytiva Life Sciences). Phosphate pH 6.0 or HBS-P+ pH 7.4 were used as running and regeneration buffers, respectively. Serial dilutions of monomeric hFcRn-His were injected over immobilized mAbs at pH 6.0 with a flow rate of 50 µl/min at 25 °C. Binding data were adjusted to a zero sample and the reference flow cell values subtracted before the Langmuir 1:1 ligand binding model (BIAevaluation software) were used to determine binding kinetics. Binding at pH 7.4 was performed by single injections of 4000 nM FcRn-His over 2000 RU of immobilized NIP IgG1 variants at 25 °C with a flow rate of 20 µl/min.

## HERA

HMEC-1 cells stably expressing HA-hFcRn-EGFP were seeded into 24-well plates (CorningCostar) at $7.5 \times 10^5$ cells/well and cultured for 2 days in complete growth medium. The cells were washed twice and starved for 1 h in Hank's Balanced Salt Solution (HBSS) (ThermoFisher). Then, 200 nM NIP specific IgG1 variants diluted in 250 µl HBSS (pH 7.4) were added in triplicates to two identical plates of cells and incubated for 4 h in a 37 °C/5% CO$_2$ incubator. The HBSS was removed, and the cells washed four times with ice cold HBSS (pH 7.4), before fresh growth medium without FCS and supplemented with MEM non-essential amino acids (ThermoFisher) was added to one of the plates and incubated ON before samples were collected (recycling fraction). Total protein lysate (residual fraction) was then isolated using RIPA lysis buffer (ThermoFisher) supplied with 1x Complete Protease Inhibitor (Roche). The mixture was incubated with the cells on ice and a shaker for 10 min followed by centrifugation for 15 min at $10000 \times g$ to remove cellular debris. Similarly, total protein lysate (uptake fraction) from cells in the second plate was isolated after 4 h using the same protocol. The amounts of NIP IgG1 variants present in the samples and lysates were quantified by ELISA.

## In vivo half-life and biodistribution

Plasma half-life experiments were performed by Jackson Laboratory Services (Bar Harbor, USA). Hemizygous FcRn transgenic mice (B6.Cg-Fcgrt$^{tm1Dcr}$ Tg(FCGRT)32Dcr/DcrJ) (Tg32) that are knockout for the mouse FcRn heavy chain (Fccgrt$^{tm1Drc}$) and express the genomic transgene of the human FcRn heavy chain (FCGRT) under the control of the human FcRn promotor (Tg32) (The Jackson Laboratory) were used to measure the plasma half-life of IgG variants and etanercept. A mix of 3 female and 2 male mice aged 7–9 weeks was used per group. The mice were pre-loaded with 500 mg/kg IVIg (privigen, CLS Behring) via i.v. administration 2 days prior to i.v. administration of the test antibodies at a dose of 5 mg/kg. Blood samples (25 µL) were drawn from the retro-orbital sinus at days 1, 2, 3, 5, 7, 10, 12, 16, 19 and 23 days post administration of the test antibodies. Blood samples were mixed with 1 µL 1% K3-EDTA to prevent coagulation and centrifuged at $17.000 \times g$ for 5 min at 4 °C. Plasma was isolated and diluted 1:10 in 50% glycerol/PBS solution before stored at −20 °C until analysis by ELISA. Half-life data were plotted as percent antibody remaining compared to the first concentration measured. Data points from the β-phase were used to calculate half-life using the formula:

$$t\frac{1}{2} = \frac{\log(0.5)}{\log\left(\frac{Ac}{Ao}\right)} x\, t \qquad (1)$$

where t$_{1/2}$ is the half-life of the antibody, A$_c$ is the amount of antibody remaining, and A$_0$ is the original amount of antibody at day 1 and t is the elapsed time[70]. Where stated NCA PK model parameters were determined from the measured antibody concentrations in plasma using gPKPDsim for MatLab[34]. Following the final plasma sample collection on day 23, the mice were overdosed with tribromoethanol. Gauge needles were inserted into the trachea of each mouse to slowly

inject 1 mL PBS into the lungs before withdrawn 2 times. BALF samples were stored at −20 °C until analysis. Half-life experiments in WT Balb/c mice (Taconic Farms) (6 mice/group, 8 weeks old) were performed at Oslo University Hospital animal facility. NIP IgG1 WT and REW were administered i.v. at 5 mg/kg and blood samples collected by puncture of the saphenous vein and collected using heparinized micro capillary pipettes at day 1, 2, 3, 4, 7, 8, 11, and 15 post injection. Half-life was calculated as above. The experiments were approved by the Norwegian Food Safety Authority.

## Immunogenicity ELISAs
96-well EIA/RIA plates were coated with anti-NIP IgG1 WT, anti-NIP IgG1 REW, anti-SARS-CoV-2 IgG1 WT and anti-SARS-CoV2 REW (mAb4) (diluted to 1 μg/mL/100 μl per well). In addition, mouse IgG from serum (Sigma-Aldrich, I5381) was coated in parallel (0.25 μg/mL) as a positive control for the detection antibody used. Then, plasma samples collected at all timepoints of the plasma half-life experiments, including pre-bleed samples, were diluted 1:200 and added to the plates followed by incubation for 1 h at RT. Captured mouse IgG was detected using an AP-conjugate anti-mouse IgG (Fc-specific from goat (Sigma-Aldrich, A9316) (diluted 1:5000 in PBS/T/S) and visualized by addition of AP-substrate (10 ug/mL in diethanolamine buffer) (Sigma-Aldrich). Absorbance values were recorded at 405 nm using a Sunrise Spectrophotometer (TECAN).

## In vivo pulmonary delivery
Homozygous B6.Cg-Fcgrttm1Dcr Tg(FcGRT)32Dcr/DcrJ mice (Tg32) (The Jackson Laboratory) (6 mice/group, 8 weeks old) were used. A mix of 3 female and 3 male mice was used per group. When sedated after intraperitoneal delivery of ZRF cocktail (250 mg/mL of Zoletil Forte, 20 mg/mL of Rompun, 50 μg/mL of Fentanyl), 10 μl of NIP IgG1 WT or REW diluted in PBS (2.23 mg/kg) was given to each nostril followed by inbreath while lying on their backs. Blood was collected by puncture of the saphenous vein and collected using heparinized micro capillary pipettes 24 h post administration and analyzed by ELISA. The experiment was performed at Oslo University Hospital animal facility and approved by the Norwegian Food Safety Authority.

## Transcytosis assays
Assays were performed using Transwell filters (1.12 cm²) with collagen coated polytetrafluoroethylene (PTFE) membranes and 0.4 μm pore size (Corning Costar). The filters were incubated ON in complete growth medium followed by seeding of $1.0 \times 10^6$ T84 or $1.4 \times 10^6$ MDCK-hFcRn cells per well. Transepithelial resistance (TEER) was monitored daily using a MILLICELL-ERS-2 volt-ohm meter (Millipore). The T84 cultures were grown for 4−5 days before reaching confluency with a TEER value of 1000−1200 Ω x cm² while MDCK-hFcRn were grown for 24 h before reaching a TEER value of 600−800Ω x cm². Prior to experiments, the cells were starved for 1 h in HBSS. Then 200 nM (200 μl) of NIP-specific IgG1 variants were added to either the apical or basolateral reservoir and incubated at 37 °C for 4 h before samples were taken from the opposite reservoir and analyzed by ELISA.

## Monovalent HA-Fc fusions
The cDNA fragment encoding amino acids 18−541 of hemagglutinin (HA) from influenza A H1N1 (A/Puerto Rico/8/1934 (PR8)) was used to generate IgG1 Fc fragments with HA fused to the N-terminal end of one of the HCs. The HA cDNA was subcloned into the pFUSE-hIgG1-Fc2 expression vector (InvivoGen) to generate pFUSE-HA(PR8)-hIgG1-Fc2. Removal of the multiple cloning sites in the target vector generated pFUSE-naked-hIgG1-Fc2. Mutagenesis was then performed to introduce the knob-in-hole mutations. The "knob" mutation (T366Y) was introduced into pFUSE-HA(PR8)-hIgG1-Fc2, while the "hole" mutation (Y407T) was introduced into the pFUSE-naked-hIgG1-Fc construct either alone or in combination with the REW substitutions. The

monovalent Fc fusions were produced in Expi293 cells by transient transfection adding a 2:1 ratio of the knob:hole constructs per the manufacturer instructions. The fusions were purified using a CaptureSelect FcXL affinity matrix (ThermoFisher) packed in a 5 ml column (Repligen) per the manufacturer recommendations. Eluted fractions were collected, concentrated and buffer exchanged to 1xPBS using Amicon Ultra-30 spin columns (Merck). Monomeric fractions of the monovalent fusions were then isolated by SEC using a Superdex 200 Increase 10/300 GL column (Cytiva Life Sciences) with an ÄKTA Avant 25 instrument (Cytiva Life Sciences).

## FcRn binding to HA-Fc fusions in ELISA
96-well EIA/RIA plates (Corning) were coated with a recombinant human albumin variant (8 μg/mL in PBS/ 100 μL per well) engineered to bind pH independently to human FcRn[29]. The plates were blocked with PBS/S and washed before recombinant soluble human FcRn-His was added and incubated at RT for 1 h. Then, stated titrated amounts of monovalent HA Fc-fusions or anti-NIP IgG1 variants were diluted in PBS/T/S pH 6.0 or pH 7.4 and added to the plates. All remaining steps were performed using PBS/T/S pH 6.0 or pH 7.4 as dilution and wash buffers. Bound monovalent HA Fc fusions or anti-NIP IgG1 variants were detected by an anti-human IgG Fc specific AP-conjugated antibody from goat (Sigma-Aldrich, A9544) (diluted 1:5000 in PBS/T/S) and visualized by addition of p-nitrophenylphosphate (Merck) (10 μg/mL in diethanolamine buffer). Absorbance values were recorded at 405 nm using a Sunrise plate-reader (TECAN).

## Mucosal vaccination and virus challenge
The experiments were performed at the Oslo University Hospital animal facility. Female homozygous B6.Cg-Fcgrttm1Dcr Tg(FcGRT)32Dcr/DcrJ mice (Tg32) aged 8−10 weeks (The Jackson Laboratory) (5 mice per group, 4 mice in the NaCl group) were used. Female mice were used due to housing considerations. At the day of vaccination, mice were anesthetized intraperitoneally with ZRF cocktail. When sedated, 10 μL of the vaccine mixtures was given in each nostril followed by inbreath while lying on their backs. Specifically, each mouse was given 20 μg CpG ODN 1826 Vaccigrade (Invivogen) mixed with 1.73 μg HA(H1N1)-Fc (WT or REW). After 3 weeks, each mouse was vaccinated in the same manner as above with 10% of the Fc fusions mixed with 20 μg CpG. The mice were challenged with a deadly dose (5x Lethal Dose 50) of *influenza A H1N1* (A/Puerto Rico/8/1934 (PR8)) after 6 weeks. Specifically, mice were anesthetized and given virus i.n. as above. Weight loss was monitored daily or every second day after infection and the endpoint was set at 20% weight loss. If the endpoint was reached, mice were terminated by cervical dislocation or $CO_2$ gas chamber. Plasma samples were harvested by saphenous vein puncture 2 weeks after the boost to measure HA antibody titers. The experiments were approved by the Norwegian Food Safety Authority.

## HA mouse antibody titer ELISA
96-well EIA/RIA plates were coated with recombinant HA (H1N1 A/PR8) protein (1 μg/mL/100 μL per well) (Sino Biological). The wells were blocked with PBS/S and washed before addition of plasma samples diluted 1:1, 1:2, 1:4, 1:8, 1:16, 1:32, and 1:64 for 1 h at RT. Bound HA specific mouse antibodies were detected using an AP-conjugated anti-mouse IgG Fc specific antibody from goat (Sigma-Aldrich, A9544) (diluted 1:4000 in PBS/T/S) and visualized by addition of AP-substrate (10 μg/mL in diethanolamine buffer). Absorbance values were measured at 405 nm using a Sunrise Spectrophotometer (TECAN). PBS/S/T (pH 7.4) or PBS/T were used as dilution and washing buffer, respectively. Antibody titer was determined as the highest dilution factor for each mouse with a higher OD value than background, where the background is the mean absorbance of mice given NaCl plus 5x the standard error of the mean of the same observations.

### Ex vivo placental perfusion model

An established ex vivo placental model was used[39,71,72]. Here, 10 μg/ml of NIP-specific IgG1 variants mixed 1:1 with infliximab (anti-TNFα, IgG1; Remicade) in a total volume of 100 ml was added to the maternal perfusion reservoirs. Samples from the fetal reservoir were collected before the test proteins were added (0 min) and from 2 min after adding the antibodies (perfusion start) followed by the time points 60, 120, 180, 210, 240, 270, 300, 330 and 360 min. Placentas from uncomplicated pregnancies resulting in vaginal birth or caesarean section were donated by women giving birth at Copenhagen University Hospital. To minimize variation, mothers who smoked, had diabetes or other pregnancy complications were excluded from the study. Only term placentas were included, and the project was approved by the ethical committees in the Communities of Copenhagen and Frederiksberg and the Danish Data Protection Agency. Informed consent was obtained in accordance with the Declaration of Helsinki.

### Quantification of IgG

Quantification of from cellular experiments, in vivo studies and the ex vivo placental perfusion model was performed using ELISA. Recombinant mAb variants were captured on either BSA-NIP(25) (1 μg/mL/100 μL per well) (Biosearch Technologies), TNF-α (1 μg/mL/100 μL per well) (Peprotech), recombinantly produced SARS-CoV-2 receptor binding domain (RBD) (Sino Biologics) (1 μg/mL/100 μL per well) or anti-human IgG Fc (Sigma-Aldrich, I2136) (1 μg/mL/100 μL per well). Captured mAb variants were detected by an AP–conjugated polyclonal anti-human IgG Fc specific Ab from goat (Sigma-Aldrich, A9544) (diluted 1:5000 in PBS/T/S) and visualized by addition of p-nitrophenylphosphate (Sigma-Aldrich) (10 μg/mL in diethanolamine buffer). Absorbance was recorded at 405 nm using a Sunrise plate-reader (TECAN). PBS/S/T (pH 7.4) or PBS/T was used as dilution and washing buffers, respectively.

### Complement ELISAs

ELISAs were performed by capturing anti-NIP IgG1 variants (3000.0–23.4 ng/mL) on BSA-NIP(16) (diluted to 1 μg/mL in PBS) (Biosearch technologies) coated in 96-well EIA/RIA plates (Corning-Costar) or by randomly coating NIP IgG1 mAbs directly in wells (10 μg/mL). For antigen density experiments BSA-NIP(3), BSA-NIP(16) or BSA-NIP(80) (Norwegian Institute of Public Health) were coated at BSA concentrations of 2.5 μg/mL, 0.5 μg/mL and 0.1 μg/mL, respectively, to obtain different surface densities with the same number of total NIP molecules. NHS (diluted 1:200 in veronal buffer) (Complement Technologies), pure human C1q (diluted to 0.366 μg/mL in veronal buffer) (Complement Technologies) or pure mouse C1q (diluted to 0.366 μg/mL in veronal buffer) (Prospecbio) was added and the plates incubated at 37 °C for 30 min. Bound C1q or deposited C3, C4 or C5 was detected using specific primary antibodies from rabbit (all from Dako/Agilent, A0135, A0062, A0065, A0056) (diluted 1:5000 in PBS/T/S) and a secondary HRP-conjugated anti-rabbit IgG antibody from donkey (Cytiva Life Sciences, NA935) (diluted 1:10.000 in PBS/T/S). Bound mouse C1q was detected by polyclonal anti-mouse C1q serum from goat (Creative Biolabs, CTA-P-023) (diluted 1:2000 in PBS/T/S) followed by detection using an anti-goat IgG AP-conjugated antibody from rabbit (Merck). Binding was visualized by addition of TMB substrate (Calbiochem) or AP-substrate (Merck) (10 μg/mL in diethanolamine buffer). The HRP reaction was terminated by adding 1 M HCl. Detection of deposited TCC (C5bC9) was done using a biotinylated mouse monoclonal antibody specific for a C9 neoepitope exposed upon C5b binding (Diatec Monoclonals, DIA 011-01) (diluted 1:5000 in PBS/T/S) together with AP-conjugated streptavidin (diluted 1:5000 in PBS/T/S) (Cytiva Life Sciences) and visualized by addition of AP substrate (Sigma-Aldrich). For the spA competition ELISA, a 5-molar excess of spA (Sigma-Aldrich) was allowed to bind captured NIP IgG1 variants

before addition of pure C1q. Absorbance values were recorded at either 450 (HRP) or 405 (AP) nm using a Sunrise plate-reader (TECAN). PBS/S was used as blocking buffer while PBS/S/T was used as dilution and washing buffers.

### Solution-phase complement activation and IgG complex formation assays

Solution phase complement assays were performed by incubating anti-NIP IgG1 WT, REW, RGY, and PGLALA in NHS (Complement Technologies) at a concentration of 100 μg/mL for 1 h at 37 °C. IgG complex formation and C4d concentrations were determined using the MicroVue CIC EIA and MicroVue C4d EIA kits (Quidel) following the manufacturer instructions.

### In vivo solution phase complement activation assays

Homozygous human FcRn transgenic mice (B6.Cg-Fcgrttm1Dcr Tg(FcGRT)32Dcr/DcrJ) (Tg32) (The Jackson Laboratory) were injected i.v. with 10 mg/kg anti-NIP IgG1 WT, REW, and RGY variants (3 mice per group, age 8–10 weeks, 2 female and 1 male per group) followed by terminal bleeding and plasma collection 24 h post-administration. In addition, plasma from 3 non-treated mice were collected. The plasma levels of anti-NIP IgG1 WT, REW and RGY were quantified by the anti-human Fc ELISA described above by interpolation to 12-point standard curves of each individual protein (1000–0.0056 μg/mL). Binding between anti-NIP IgGs and mouse C1q in the plasma samples was measured by coating anti-mouse C1q serum from goat (Creative Biolabs, CTA-P-023) (diluted 1:500 in PBS) in 96-well EIA/RIA plates followed by blocking for 2 h ar RT. Then human IgG concentration normalized plasma samples were added and incubated for 1 h at RT. Bound human IgG1 variants were detected by a goat anti-human IgG Fc specific AP-conjugated antibody (Sigma-Aldrich A5944) (diluted 1:5000 in PBS), and binding visualized by addition of AP-substrate (10 μg/mL in diethanolamine buffer) (Sigma-Aldrich). Absorbance values were recorded at 405 nm using a Sunrise Spectrophotometer (TECAN). PBS/S was used as blocking buffer, PBS/S/T was used as dilution buffer and PBS/T was used as wash buffer in between each step. Levels of mouse C3a in human IgG concentration normalized plasma samples were measured using TECO Mouse C3a assay kit (Quidel) following the manufacturer's instructions.

### CDC Calcein-AM release assay

Raji, WSU-NHL, DOHH2, SU-DHL-4 cells were washed in 10 mL HBSS and resuspended to $1.0 \times 10^7$ cells/mL in HBSS before stained with 1 μM Calcein-AM (Merck) for 20 min at RT. The cells were then washed twice with 10 mL HBSS and diluted in HBSS to a density of $1.0 \times 10^6$ cells/mL before 50 μL was added to 96-well V-bottom plates ($5.0 \times 10^4$ cells/well) together with 25 μL NHS (Complement Technologies) (25% final concentration) and 25 μl anti-CD20 IgG1 variants (0.74 μg/mL final concentration). 50 μL RIPA buffer (ThermoFisher) instead of antibodies and NHS was used to determine maximum lysis of the target cells. Control samples containing 25 μL HBSS + 50 μL cells and 25 μL NHS were included to account for background. The plates were incubated at 37 °C/5% CO₂ for 1 h before centrifuged at $1314 \times g$ for 5 min. 50 μl supernatant was then transferred to black clear bottom optical 96-well Viewplates (Perkin Elmer) and fluorescent intensity was determined at 485 nm excitation/510 nm emission using an Envision plate reader (Perkin Elmer). Percent antibody-mediated lysis was calculated relative to the max lysis control.

### Phagocytosis of *S. aureus* by PMNs

GFP-expressing Newman Δspa/sbi or Newman WT ($7.5 \times 10^5$ CFU) was incubated with human monoclonal anti-WTA (4497) IgG1 WT, REW, or PGLALA, and 1% ΔIgG/IgM serum in RPMI + 0.05% HSA (RPMI-HSA), for 15 min at 37 °C with shaking (±700 rpm), in a round-bottom microplate. Bacteria were then incubated with freshly isolated human

neutrophils ($7.5 \times 10^4$), that were purified from blood of healthy donors by the Ficoll/Histopaque density gradient method[73], for another 15 min at 37 °C with shaking. All samples were fixed with 1% paraformaldehyde in RPMI-HSA. The binding/internalization of GFP bacteria to the neutrophils was detected using flow cytometry (BD FACSVerse), and data were analyzed based on forward/side scatter gating of neutrophils using FlowJo software. The use of human neutrophils was approved by Ethics Committee NedMec under informed consent from healthy donors.

### Phagocytosis of *S. pneumoniae* by PMNs

Human neutrophils were freshly isolated from healthy donor blood using the Ficoll-Histopaque gradient method[74]. Phagocytosis assay was performed in a round-bottom 96-well plate and neutrophil-associated fluorescent bacteria were analyzed by flow cytometry. FITC-labeled *S. pneumoniae* were opsonized by pre-incubation with 2-fold serial dilutions of the antibodies in IgG/IgM-depleted serum[75] as complement source, in RPMI-HSA for 20 min at 37 °C. Subsequently, neutrophils were added in a 1:10 cell to bacteria ratio and phagocytosis was allowed for 15 (*S. aureus*) or 30 min (*S. pneumoniae*) at 37 °C on a shaker (650 rpm). Ice-cold 1% PFA in RPMI-HSA was added to stop the reaction. Samples were measured by flow cytometry, and % of positive cells and mean fluorescence values were determined for gated neutrophils[74]. The use of human neutrophils was approved by Ethics Committee NedMec under informed consent from healthy donors.

### *N. gonorrhea* CDC assay

Bacteria harvested from ON cultures were re-passaged on chocolate agar, grown for 6 h, and suspended in HBSS containing 0.15 mM CaCl$_2$ and 1 mM MgCl$_2$ (HBSS$^{++}$). About 2.000 CFU of suspended bacteria were incubated with human complement (IgG and IgM depleted normal human serum; Pel-Freez) and titrated amounts of 2C7 IgG variants. The final reaction volumes were maintained at 150 µL. Aliquots of 25 µL of the reaction mixtures were plated onto chocolate agar in duplicates at the beginning of the assay (t$_0$) and again after incubation at 37 °C for 30 min (t$_{30}$). Survival was calculated as the number of CFU´s at t$_{30}$ relative to t$_0$. Binding of Humab 2C7 variants to the surface of *N. gonorrhoeae* was measured performed by flow cytometry[76].

### FcγR binding ELISAs and SPR

Anti-NIP IgG1 variants (2 µg/mL in PBS) were captured on BSA-NIP(25) (1 µg/mL in PBS) (Biosearch Technologies) coated in 96-well EIA/RIA plates (CorningCostar) for 1 h at RT. Biotinylated human FcγRI, FcγRIIa-H131, FcγRIIa-R131, FcγRIIb, FcγRIIIA-V158 and FcγRIIA-F158 (10 µg/mL) (Sino Biological) were then added and incubated for 1 h at RT. Bound receptors were detected with streptavidin-AP (diluted 1:500 in PBS/T/S) (Roche Diagnostics). Biotinylated FcγRIIIb (1 µg/mL in PBS) was pre-incubated with streptavidin-AP to increase the detection sensitivity. The 405 nm absorption spectrum was recorded using a Sunrise plate-reader (TECAN). PBS/S (pH 7.4) was used as blocking buffer while PBS/S/T (pH 7.4) and PBS/T was used as dilution and washing buffers, respectively. A Biacore T200 instrument (Cytiva Life Sciences) was used to obtain SPR binding curves of NIP IgG1 variants to biotinylated human FcγRI, FcγRIIa-H131, FcγRIIa-R131, FcγRIIb, FcγRIIIA-V158 and FcγRIIA-F158 (Sino Biological). The receptors were captured on Series S SA chips at 200 RU and binding to the low affinity FcγRs recorded by single injections of 100 nM anti-NIP IgG1 variants at a flow rate of 10 µL/min and a contact time of 60 s. For high affinity FcγRI, 50 nM anti-NIP IgG1 was injected at a flow rate of 40 µL/min with a contact time of 30 s. The maximum binding responses were normalized to 100 RU using the BIAevaluation software to allow overlay of the sensorgrams.

### ADCC

ADCC was performed by an [$^{51}$Cr] release assay[50]. MNCs, anti-CD20 IgG1 variants (0.01–100 nM) and medium were added to round-bottom microtiter plates (Nunc, Rochester, NY, USA). Assays were started by addition of effector and target cells at E:T ratio 40:1 (200000:5000). After 3 h at 37 °C, [$^{51}$Cr] release from 5 parallel samples was measured. Percentage of [$^{51}$Cr] release was calculated using the formula: % lysis = (experimental cpm − basal cpm)/(maximal cpm − basal cpm) × 100; maximal [$^{51}$Cr] release was determined by adding Triton-X (2% final concentration) to target cells, and basal release measured in the absence of antibodies. The use of human MNCs were approved by the Ethics Committee of Kiel University under informed consent from healthy donors.

### ADCP

Monocyte derived macrophages were generated by letting them attach to the cell culture flask for 30 min at 37 °C in monocyte-attachment medium (PromoCell). The monocyte attachment medium was then removed, and the cell culture flasks were washed with 1xPBS before the monocytes were cultured in X-VIVO™ 15 media (Lonza) with 25 µg/mL streptomycin and 25 U/mL penicillin (ThermoFisher). For macrophage generation 50 ng/mL M-CSF (PeproTech) was added every 3 days for at least 9 days. Real time, automated ADCP was measured by fluorescence microscopy using Incucyte® (Sartorius). Target cells were labeled with 0.5 µg/mL pHrodo dye (ThermoFisher) for 1 h at room temperature. M-CSF derived macrophages were added as effector cells using an effector/target cell ratio of 1:1 (40.000:40.000). The assay was started by adding anti-CD20 mAb2 antibody variants (100 nM). Then, ADCP was measured for 10 h every 10 min at 37 °C. ADCP was determined as red counts per image, which correlate to the number of tumor cells engulfed by the macrophages. Analysis was performed using Top-Hat segmentation, 2 red calibrated units (RCU) as threshold and 20 µm radius. The minimum intensity was set to 8–8.5. Under these conditions control samples (pHrodo cells only) started at ≤10 counts at time point 0. The use of human MNCs was approved by the Ethics Committee of Kiel University under informed consent from healthy donors.

### CDC $^{51}$Cr release assay

An established CDC $^{51}$Cr assay was used[77]. Target cells per condition were labeled with 200 µCi [$^{51}$Cr] for 2 h. To the Raji target cells ($1.0 \times 10^4$ per condition), 25% v/v freshly drawn human serum was added as source of complement in the presence of anti-CD20 IgG1 variants (0.01–100 nM). The percentage of cellular cytotoxicity was calculated using the same formula as for $_{51}$Cr ADCC assays. The use of human MNCs were approved by the Ethics Committee of Kiel University under informed consent from healthy donors.

### SDS-PAGE

Protein samples (2 µg) were analyzed on 12% Bis-Tris Bolt SDS-PAGE gels (ThermoFisher) with Bolt MES SDS running buffer (ThermoFisher). A Spectra Multicolor Broad Range Protein Ladder (ThermoFisher) was used as standard, and the gels were stained with Comassie Brilliant Blue (BioRad).

### Analytical SEC

Analytical SEC was performed by applying 20 µg of each anti-NIP IgG variant to a Superdex 200 Increase 3.2/300 analytical SEC column (Cytiva Lifesciences) at a flow rate of 0.05 mL/min using an AKTA FPLC instrument (Cytiva Life sciences). Data was normalized to relative fluorescence for clarity.

### Differential scanning fluorimetry (DSF)

DSF was performed either by a dye-based method using a Lightcycler RT-PCR instrument (Roche) or label-free using a Prometheus NT.48 nanoDSF instrument (Nanotemper Technologies GmbH). For dye-based DSF, SYPO Orange (Sigma-Aldrich) was used at a dilution of 1:1000 with a protein concentration 0.1 mg/mL in a final volume of

25 μL. All samples were run in triplicates in 96-well Lightcycler 480 multiwell plates. The peaks of excitation and emission filters were used and the instrument programed to raise the temperature from 25 °C to 95 °C after a stabilization time of 10 min at 25 °C. Data was collected every 0.5 °C. Data transformation and analysis were performed using a DSF analysis protocol[78]. For label-free DSF, 1 mg/mL samples were drawn into capillaries in triplicates. The instrument was set to gradually increase the temperature (2 °C/min) from 25 °C to 95 °C. The melting temperature (Tm °C) in which half of the proteins were unfolded was determined by deducing the first derivative in the PR. ThermoControl software.

### Rheumatoid factor ELISA

Recombinantly produced IgG1 WT and REW Fc fragments were coated (1 μg/mL/100 uL per well) in 96-well EIA/RIA plates (CorningCostar) for 1 h at RT. Then Rh+ human serum (Lee BioSolutions) was diluted 1:1–1:10^6-fold in PBS, added to the plates and incubated for 1 h at RT. NHS was added at 1:1 dilution as a negative control. The plates were washed 4 times with PBS/T/S. Serum antibodies bound to the coated Fc fragments were detected using a pan anti-human IgG light chain antibody from rabbit (ReMab Biosciences, 32-1031-00) (diluted 1:3000 in PBS/T/S) and visualized by addition of anti-rabbit IgG-HRP (Cytiva Life Sciences, NA935) (diluted 1:3000 in PBS/T/S). The coating efficacy of WT and REW Fc fragments were controlled using an AP conjugated polyclonal anti-human IgG (Fc specific) antibody from goat (Sigma-Aldrich, A5944) (diluted 1:5000 in PBS/T/S). The 450 nm or 405 nm absorbance values were recorded using a Sunrise plate reader (Tecan).

### Analytical FcRn affinity chromatography

was performed using a human FcRn retention column and a ÄKTA Avant 25 instrument (Cytiva Life Sciences). 100 μg NIP IgG1 variants were injected over the column and allowed to bind FcRn at pH 6.0 before being subjected to an increasingly more basic pH gradient (pH 6.0 to 8.8) over 110 min by mixing of two eluent buffers (20 mM MES sodium salt, 140 mM NaCl, pH 6.0 and 20 mM Tris/HCl, 140 NaCl, pH 8.8). The pH was continuously recorded by a pH detector (Cytiva Life Sciences).

### Protein A ELISA

Titrated amounts of NIP IgG1 WT or REW (1000.0–0.78 ng/mL) was captured on BSA-NIP(25) (diluted to 1 μg/mL in PBS) coated in 96-well EIA/RIA plates (CorningCostar) pre-blocked with PBS/S. Then AP conjugated Protein A from *S. aureus* (Sigma-Aldrich) was diluted 1:5000 in PBS/T/S before visualization of binding by addition of AP substrate (diluted to 10 μg/mL in diethanolamine). The 405 nm absorption values were recorded using a Sunrise spectrophotometer (TECAN).

### Liquid chromatography tandem mass spectrometry analysis

Liquid chromatography tandem mass spectrometry was performed by mixing 50 μl of each IgG1 variant (1 mg/mL) with 1 μg trypsin dissolved in 100 μl 50 mM ammonium bicarbonate (pH 7.8) and incubated ON at 37 °C. Peptides were isolated by collecting the flow-through from centrifugal filters and transferred to Eppendorf tubes before drying using SpeedVac (HetMaxi dry). Dried samples were dissolved in 20 μl 0.1% formic acid, sonicated for 30 s and centrifuged for 10 min at 16,100 × g. 10 μl of each samples was transferred to new vials, and reverse phase (C18) nano outline liquid chromatography-tandem mass spectrometry (LC-MS/MS) analysis of proteolytic peptides was performed using a system of two Agilent 1200 HPLC binary pumps (nano and capillary) with an autosampler, column heater and integrated switching valve. The system was coupled via a nanoelectrospray ion spource to an LTQ Orbitrap mass spectrometer (ThermoFisher). For the analysis 6 μl peptide solution was injected into the 5 × 0.3 mm extraction column filled with Zorbax 300SB-C18 of 5 μm particle size (Agilent Technologies). After

washing for 5 min with 0.1% formic acid (v/v) and 3% acetonitrile (v/v) at a flow rate of 10 μl/min, the integrated switching valve was activated and peptides were eluted in the back-flush mode from the extraction column onto a 150 × 0.075 mm C18, 2 μm resin column (GlycoproSIL C18-80Å, Glycopromass). The mobile phase consisted of acetonitrile and mass spectroscopy-grade water both containing 0.1% formic acid. Chromatographic separation was achieved using a binary gradient from 5 to 55% of acetonitrile in water for 1 h with a flow rate of 0.2 μl/min. Mass spectra were acquired in the positive ion mode applying a data-dependent automatic switch between survey scan and MS/MS acquisition. Peptide samples were analyzed with a high-energy collisional dissociation (HCD) fragmentation method with normalized collision energy at 25 and 41, acquiring one Orbitrap survey scan in the mass range of m/z 300–2000 followed by MS/MS of the three most intense ions in the Orbitrap (R7500). The target value in the LTQ-Orbitrap was 1 million for survey scan at a resolution of 30,000 at m/z 400 using lock masses for recalibration to improve the mass accuracy of precursor ions. Ion selection threshold was 500 counts. Selected sequenced ions were dynamically excluded for 180 s. Data analysis was performed on Xcalibur v2.0. MS/MS spectra for all glycopeptides and these were extracted by oxonium ion search; 204.086 (*N*-acetylhexosamine) and 366.1388 (*N*-acetylhexosamine-hexose) were used. HCD fragmentation with normalized collision energy at 25 was used to detect the glycans, and the peptide mass was detected for the IgG glycopeptides. Extracted ion chromatogram for target glycopeptides (EEQYNSTYR and the miscleaved TKPREEQYNSTYR with all different glycan masses) were extracted with 10 ppm accuracy and MS/MS spectra were manually verified. HCD fragmentation with normalized collision energy at 41 was used to detect the peptide sequence and to verify that the peptide mass corresponded to the correct peptide sequence. The area under the curve for all extracted glycopeptides was calculated and the percentage ratio for each glycoform was determined.

### Immunogenicity prediction

Prediction of T-cell epitopes was performed using NetMHC4.1[79] with 9mer peptides against representative human HLA supertypes using the default rank thresholds for strong (0.5) and weak (2.0) binders. B-cell epitope prediction was performed using the Immune Database analysis resource antibody epitope prediction tool[80] with a threshold of 0.5.

### Statistical analysis

Figures and statistical analyses were prepared using GraphPad Prism (GraphPad Software).

### Reporting summary

Further information on research design is available in the Nature Portfolio Reporting Summary linked to this article.

## Data availability

All data supporting the findings described are available in this paper and in the Supplementary Information. The Source Data and Supplementary Information are provided with this paper. Source data are provided with this paper.

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

## Acknowledgements

This work was partially supported by the Research Council of Norway through its Centers of Excellence scheme, project number 332727, the Global Health and vaccination research (GLOBVAC) program, project 285136 (J.T.A., S.F., D.S., J.B., S.A.S.), the grants 274993, 287927 (J.T.A. and F.R.J.) and 335688 (S.F.), the South-Eastern Norway Regional Health

Authority project 2018052 (J.T.A.), the Norwegian Cancer Society, Grant no. 223315 (J.T.A. and S.F.) as well as the Novo Nordisk Distinguished Innovator Grant NNF22OC0076567. The project also received funding from European Research Council (ERC) under the European Union's Horizon 2020 research and innovation program (grant agreement No. 101001937, ERC-ACCENT to S.H.M.R.). M.L. and T.V. were supported by the German Research Organization (DFG; CRU 5010, P6). J.B. was supported by DBT Wellcome Trust India Alliance Team Science Grant (IA/TSG/19/1/600019).

## Author contributions

S.F. and J.T.A., designed research, S.F., S.A.S., L.A., M.L., J.S., A.R.C., L.S., L.M., F.R.J., A.K.A., T.T.G., S.M. and M.B. performed research and analyzed data, M.E., D.B.B., T.E.M., T.S., D.S., J.B., J.L., T.V., S.R., S.H.M.R. and I.S., provided resources, S.F. and J.T.A. wrote the paper and prepared figures. All authors revised and approved the final version of the manuscript.

## Competing interests

J.T.A., I.S. and S.F. are inventors of the invention claimed in a patent family arising from WO2017158426 and WO2019057564 with the title "Engineered immunoglobulins with altered FcRn binding". The remaining authors declare no competing interests.
