## [Peer Review File · Nature Communications]

Human IgG Fc-engineering for enhanced plasma half-life, mucosal distribution and killing of cancer cells and bacteriaREVIEWER COMMENTS

Reviewer #1 (Remarks to the Author):

In the present study, Foss et. al. outline a novel engineered Fc containing three single amino acid substitutions intended to increase the affinity of the Fc for the human neonatal Fc receptor. The combination of in vitro and in vivo assays that the authors use to characterize their engineered Fc clearly demonstrate its improvement compared to WT Fc controls in human FcRn binding and serum half-life. Additionally, the results suggested that the REW variant may improve on-target complement activation and likely does not interfere with either afucosylation-dependent ADCC or subclass-specific activation of complement pathways, though further characterization of the REW variant in different IgG subclasses may be of interest. Overall, the REW Fc variant appears to be a potentially useful tool for the development of engineered Fc monoclonals and Fc fusion proteins. However, there are a few limitations that the authors need to address to further characterize the REW variant.

1. It appears that the reported half-life (even for wild-type IgG1) is extremely long (much longer than it was previously reported). Also, despite the fact that mouse FcRn has increased affinity for human IgG1 compared to human FcRn, wild-type mice (Balb/c) showed reduced half-life for WT IgG1 compared to hFcRn Tg mice. How do the authors explain this?
2. Related to #1, what was the rationale for including a pre-dosing step of IVIg for all the half-life studies? Mice already have endogenous IgGs, which can interact with sufficient affinity with human FcRn. The inclusion of additional human IgGs is actually expected to have a negative impact on the half-life of human mAbs (wt or REW variants) due to competition for FcRn.
3. Although there are some studies on ADCC and ELISA data on FcRIIIa binding, a more comprehensive characterization (by SPR) is necessary to assess the affinity of the REW variant (along with FcRn-enhancing variants, like LS and YTE as controls) against all classes of human FcRs.
4. The studies on the immunogenicity of HA-Fc fusions need additional characterization. In the current form, they don't offer anything to the main conclusion of the study. There's need for a more thorough characterization of the breadth and potency of the elicited IgG responses, titer analysis, passive transfer studies etc. Also, it is not clear whether the observed effects are due to FcRn enhancement or due to increased complement activation. Studies in KO mice, as well as the inclusion of several additional experimental controls are necessary.
5. Have the authors assessed the immunogenicity of the Fc variants and if so, how do they explain the lack of it? Previous studies were performed in either Rag-KO or hIgG1-Tg mice, so immunogenicity has never been an issue. But, since this study has been performed in fully immunocompetent mice, why there are no anti-human IgG responses, especially since mice were treated with such a high dose of IVig, as well as since the authors were able to detect human IgGs for several weeks post-administration?
6. In Figure S3I, why there is no difference in FcRn binding between WT and REW Fc fusions of HA?

Reviewer #2 (Remarks to the Author):

The manuscript by Foss and colleagues addresses a timely and highly interesting topic. Identifying IgG variants with optimized effector functions and tissue distribution is key to achieve better clinical responses in treating cancer, autoimmunity as well as infectious diseases. The study represents a tour de force, starting with the identification of an IgG variant (REW) which better FcRn binding based on crystal structure data, characterization of the functional properties of this variant (complement activation, cancer cell killing in vitro, etc.), to ultimately testing the enhanced function of this variant in in vivo model systems of influenza infection. In general, the study is performed at very high standards, contains clearly present and interpretable results and demonstrates that the REW IgG variant may have great potential.

Suggestions to further improve the impact of the study:

The influenza vaccination study is impressive and suggests that the HA-IgG1REW construct induces a stronger protective immune response. Do the authors have further immunologic data to support this very interesting result (anti-HA antibody levels or enhanced HA specific cytotoxic T cells)?

The authors may consider to include a syngeneic lung tumor model (B16F10 or alike) where cytotoxic antibodies are available. With respect to B16F10 the TA99 antibody could be generated on a human IgG1 format including the REW mutations. Human IgG1 binds well to mouse FcRIV and FcRI and thus should have enhanced activity against lung tumors. This type of experiment is only possible of course if the REW mutant retains binding to mouse FcRs I and IV.

The authors correctly note that the REW variant should not result in spontaneous complement activation in the serum. Thus it would be great to show that in vivo no spontaneous activation of the complement system occurs. As before it would need to be assessed if mouse C1q binds to REW in a comparable fashion to human C1q. If it does, this would be a highly interesting set of data.

As the REW variant shows increased FcRn binding it would be of great interest to study at which concentration or if at all it behaves as a bona fide FcRn blocking agent. This could be studied by looking at enhanced clearance of IVIg or human serum albumin compared to untreated or efgartigimod (or any other FcRn blocker).

November 27th, 2023

Point-by-point response to reviewers: NCOMMS-23-15179-T.

Reviewer #1 (Remarks to the Author):

In the present study, Foss et. al. outline a novel engineered Fc containing three single amino acid substitutions intended to increase the affinity of the Fc for the human neonatal Fc receptor. The combination of in vitro and in vivo assays that the authors use to characterize their engineered Fc clearly demonstrate its improvement compared to WT Fc controls in human FcRn binding and serum half-life. Additionally, the results suggested that the REW variant may improve on-target complement activation and likely does not interfere with either afucosylation-dependent ADCC or subclass-specific activation of complement pathways, though further characterization of the REW variant in different IgG subclasses may be of interest. Overall, the REW Fc variant appears to be a potentially useful tool for the development of engineered Fc monoclonals and Fc fusion proteins. However, there are a few limitations that the authors need to address to further characterize the REW variant.

1. It appears that the reported half-life (even for wild-type IgG1) is extremely long (much longer than it was previously reported). Also, despite the fact that mouse FcRn has increased affinity for human IgG1 compared to human FcRn, wild-type mice (Balb/c) showed reduced half-life for WT IgG1 compared to hFcRn Tg mice. How do the authors explain this?

We thank the reviewer for raising this question, which relates to experimental consistency and the importance of considering distinct FcRn-IgG cross-species binding differences when performing PK studies of human IgG antibodies and Fc-engineered variants in human FcRn transgenic mouse models (not expressing mouse FcRn) as well as in wild-type mice.

It is unclear to us which data the reviewer is referring to when stating that the reported half-life of wild-type IgG1 is extremely long compared to previously reported data. The plasma half-life of NIP-IgG1-WT reported in our manuscript using human FcRn hemizygous transgenic mice (Tg32: JAX) was 7.8 ± 1.1 days. This is within the range that we have previously reported for the same antibody using the same mouse strain: 8.4 ± 2.5 days (Grevys, Nat Com, 2018) and 8.5 ± 1.0 days (Gjølberg, Commun Biol, 2022).

In this regard it is important to note that while natural IgG1 antibodies have an average plasma half-life of 3 weeks in humans, the plasma half-lives of monoclonal IgG1 antibodies can vary considerably depending on the biophysical properties of their variable regions (Schoch, PNAS, 2015; Wang, Drug Metab Dispos, 2011; Schlothauer, mAbs, 2013). In fact, the plasma half-life of monoclonal IgG1 antibodies with distinct variable regions will also vary in human FcRn expressing mice, as demonstrated for the different monoclonal IgG1 antibodies used in our study.

The half-life of human IgG variants will vary between wild-type mice and human FcRn transgenic mice due to distinct differences in the ability to engage the mouse and human forms of FcRn as a function of pH. It is well documented by us and others that mouse FcRn has increased affinity at acidic pH for human IgG1. While it still binds pH dependently, it has increased binding at neutral pH (Ober, Int Immunol, 2001; Zhou, J Mol Biol, 2003; Zhou, J Mol Biol, 2005; Vaccaro, PNAS, 2006; Andersen, J Biol Chem, 2010). This is also demonstrated in the mouse FcRn binding ELISA in our manuscript (Fig. S3m). Importantly, efficient FcRn mediated recycling is dependent on uptake by fluid-phase pinocytosis followed by FcRn-engagement of IgG in acidified endosomes (5.5-6.0) that as a complex is subsequently trafficked back to the cell surface where the ligand is released from the receptor back into circulation upon exposure to the physiological pH

of the blood (7.4). As such, pH dependent binding is required for efficient rescue from intracellular degradation, and modulation of this mode-of-binding may result in either increased or decreased ability to be released from the receptor during exocytosis.

In the case of a typical wild-type human IgG1 antibody, a longer plasma half-life in wild-type mice is expected compared to that measured in human FcRn transgenic mice. This has been exemplified by the Roopenian lab that directly compared mouse IgG1 with human IgG1 having the same variable regions in wild-type and human FcRn transgenic mouse strains (Petkova, *Int Immunol*, 2006). In our study we observed a shorter plasma half-life of NIP-IgG1-WT in wild-type mice (Balb/c) (4.6 ± 0.4 days) than in human FcRn transgenic mice (B6) pre-loaded with IVIg (7.8 ± 1.1 days). We argue that these data are not directly comparable due to the different mouse strains used as well as the differences in competitive pressure on the receptor from pooled human IgG (IVIg) in human FcRn transgenic mice and endogenous mouse IgG in wild-type mice (which relates to question 2 below). Our results are in line with previously published data on the same antibody in Balb/c mice: 4.9 ± 0.4 days (Grevys, *Nat Com*, 2018).

As discussed above, too strong binding between IgG and FcRn at neutral pH reduces the ability of IgG to be released from the receptor during exocytosis. This has been demonstrated for Fc-engineered human IgG1 variants where the potential gain in plasma half-life due to increased receptor binding at acidic pH can be offset by increased binding at neutral pH (Vaccaro, *Nat Biotechnol*, 2006; Yeung, *J Biol Chem*, 2009; Borrok, *J Biol Chem*, 2015). This is extremely important regarding Fc-engineering strategies for half-life extension of human IgG1 variants. To our knowledge, all human IgG1 Fc-engineered variants reported with improved pH dependent binding to human FcRn have been shown to also improve binding to mouse FcRn, but importantly, disrupt pH dependent binding (Grevys, *Nat Com*, 2016). Consequently, such human IgG Fc-engineered variants result in shorter plasma half-life than the wild-type human IgG counterpart in wild-type mice (Dall'Acqua, *J Biol Chem*, 2002; Vaccaro, *PNAS*, 2005; Grevys, *Nat Com*, 2018). This is also the case for REW, as reported in our manuscript (Fig. S3n).

Therefore, wild-type mice (expressing mouse FcRn) are not reliable *in vivo* models for pre-clinical measurements of Fc-engineered human IgG1 candidates, including REW. This is an important aspect to consider when planning studies in disease specific mouse models.

Text has been added to the result section to clarify this point (186-189).

2. Related to #1, what was the rationale for including a pre-dosing step of IVIg for all the half-life studies? Mice already have endogenous IgGs, which can interact with sufficient affinity with human FcRn. The inclusion of additional human IgGs is actually expected to have a negative impact on the half-life of human mAbs (wt or REW variants) due to competition for FcRn.

We thank the reviewer for raising this question, which relates to the fact that studies in mice are performed under pathogen-free housing and that there are cross-species binding differences that must be considered when using human FcRn transgenic mice to explore IgG Fc-engineered variants for plasma half-life extension.

It is well documented by us and others that mouse IgG does NOT interact with human FcRn, except for very weak binding of mIgG2b (Ober, *Int Immunol*, 2001; Andersen, *J Biol Chem*, 2010). This, combined with pathogen-free housing, result in very low levels of endogenous circulating IgG in human FcRn transgenic

mice. In fact, the human FcRn transgenic mice used in our study have comparable levels of endogenous mouse IgG to that of mice engineered to lack expression of the receptor (Tam, mAbs, 2013).

The low plasma levels of mouse IgG are in stark contrast to the situation in humans with about 10-12 mg/ml of endogenous IgG that exerts a significant competitive pressure on FcRn binding. The consequence of this is that the favorable feature of Fc-engineering may be masked due to lack of competition for FcRn. To compensate for this, the mice can be pre-loaded with pooled human IgG (IVIg) to mimic natural competition for FcRn *in vivo* (Petkova, Int Immunol, 2006; Tam, mAbs, 2013; Low, mAbs, 2020). By introducing competition, both wild-type IgG and Fc-engineered variants may have shorter half-lives, but less so for an Fc-engineered variant with favorable ability to engage the receptor in the presence of competition. Hence, this trick will allow a clearer differentiation between IgG variants with distinct ability to bind human FcRn, a situation that more closely resemble a human setting where competition is present.

Text has been added to the result section (line 167-169) to clarify this point.

To demonstrate this in the context of REW, we have performed a plasma clearance experiment in which anti-NIP IgG1-WT and IgG1-REW were administered to human FcRn hemizygous transgenic mice at 5 mg/kg without IVIg pre-loading. The results showed that the plasma half-life of both antibodies was longer than in the presence of competition but also that the difference between the variants was less pronounced. These results have been added to Fig. S3g, and related text to the result (line 179-181) and method sections (line 483-484).

3. Although there are some studies on ADCC and ELISA data on FcRIIIa binding, a more comprehensive characterization (by SPR) is necessary to assess the affinity of the REW variant (along with FcRn-enhancing variants, like LS and YTE as controls) against all classes of human FcRs.

We thank the referee for this suggestion and agree. To this end, we have performed ELISA set-ups where the anti-NIP IgG1 WT, REW, LS, YTE and DHS variants were captured on the cognate antigen followed by adding soluble recombinant forms of all human FcγRs. The results showed that the antibody variants bound the high affinity FcγRI with slightly increased binding of REW and slightly reduced binding for YTE. Further, REW and LS showed increased binding to the low affinity FcγRs, DHS comparable binding to that of the WT while YTE showed reduced binding, where the latter agrees with published data (Ko, Nature, 2014; Grevys, J Immunol, 2015).

To determine whether the human FcγR binding differences between anti-NIP IgG1 WT and REW were an effect of antigen capture and potential formation of Fc:Fc contacts that could influence avidity, we performed SPR in the absence of cognate antigen. To do so, we captured site-specifically biotinylated human FcγRs on streptavidin immobilized SPR chip followed by injection of equal amounts of the two antibodies. The results showed very similar binding profiles for IgG1 WT and REW toward all the receptors. For REW, a slightly slower on-rate and off-rate were observed for FcγRIIIa-H131 and FcγRIIIa-R131 and a slightly slower on-rate and faster off-rate were observed for FcγRI. Thus, the data suggest that Fc-engineering may slightly affect engagement of the FcγRs which will depend on the context, with or without the present of antigen.

The results have been added to a new Fig. S10, and a description of the data in SI Text 5 (line 1544-1550), which are referred to in the manuscript (line 353-355) and the method section has been updated (line 694-697 and line 699-706).

4. The studies on the immunogenicity of HA-Fc fusions need additional characterization. In the current form, they don't offer anything to the main conclusion of the study. There's need for a more thorough characterization of the breadth and potency of the elicited IgG responses, titer analysis, passive transfer studies etc. Also, it is not clear whether the observed effects are due to FcRn enhancement or due to increased complement activation. Studies in KO mice, as well as the inclusion of several additional experimental controls are necessary.

We thank the reviewer for this question. Based on our *in vitro* and *in vivo* experiments showing enhanced transcellular transport of an REW-containing IgG1 compared to the WT, we were motivated to address the benefit of REW for an Fc-fusion design. To do so, we choose to include a vaccine example where the globular domain of HA from H1N1 was fused to IgG1-Fc WT or REW. The formats were used for intranasal delivery in the presence of adjuvant. In line with improved human FcRn binding of the REW containing variant, we measured a more protective response compared with the WT fusion counterpart upon challenge with a deadly dose of the virus. We agree with the referee that this is a highly interesting finding, which deserves further investigation regarding the mechanism of action and the immune responses induced upon vaccination. While we measured complete protection, there are some limitations to the experimental set-up that are discussed below.

To partly answer the question, we have used the plasma samples collected from the mice to measure the presence of anti-HA IgG antibodies using an established ELISA. The results showed a modest increase in anti-HA specific antibodies in mice vaccinated with HA-Fc-REW compared to HA-Fc-WT while no HA specific antibodies were detected in plasma from mock treated (NaCl) animals. While these data are indicative, we would like to highlight that the levels of detected antibodies were low, which is in line with the limitation described above related to the fact that human FcRn expressing mice do not rescue endogenous IgG from degradation. As such, while the mice are responding to the vaccine by generating anti-HA IgG antibodies, these are not rescued by FcRn-mediated recycling (short half-life).

The new data has been added to Fig. S3v and described in the result section (line 212-213) as well as in the method section (line 570-571 and 574-583).

As the above biology hampers extensive studies on the material at hands, it will require large set-ups for *in vivo* studies to be run over an extended time to gain an in-depth understanding, which we argue is beyond the scope of the current manuscript focusing on the design of the REW technology and its versatility. However, as we share the enthusiasm by the referee, we have initiated a project where we are exploring this finding further, but then in the context of a new mouse model expressing both human FcRn and human IgG1-Fc (Low, mAbs, 2020). In addition, we are in this context also exploring a new vaccine design that we cannot to disclose due to an on-going process related to intellectual property rights. As we have data supporting that this modified strategy is even better, supported by experiments performed in a more optimal double transgenic mouse model, we are not permitted by our animal facility to perform any additional experiments with the setup included in the current manuscript due to ethical reasons (the 3Rs). We sincerely hope that the referee understands this situation and can agree that the on-going work is beyond the main scope of the manuscript.

We have added additional text to the discussion (line 362-367) to emphasize the need for further studies on FcRn-targeted non-invasive delivery of Fc fusions, including subunit vaccines.

5. Have the authors assessed the immunogenicity of the Fc variants and if so, how do they explain the lack of it? Previous studies were performed in either Rag-KO or hlgG1-Tg mice, so immunogenicity has never been an issue. But, since this study has been performed in fully immunocompetent mice, why there are no anti-human IgG responses, especially since mice were treated with such a high dose of IVig, as well as since the authors were able to detect human IgGs for several weeks post-administration?

We fully agree with the reviewer that the potential to induce immunogenicity is important to assess. However, we do not agree that the majority of previous pre-clinical PK evaluations of human monoclonal IgG1 candidates and Fc engineered variants have been performed in Rag-KO or hlgG1-Tg mice, as to our knowledge, such studies are routinely performed in immunocompetent human FcRn transgenic mice in which the antibodies are usually well tolerated and display monophasic beta phase clearance kinetics (Zalevsky, Nat Biotechnol, 2010; Avery, mAbs, 2019; Grevys, Nat Com, 2016; Gjølborg, Commun Biol, 2022; Schoch; PNAS, 2015; Ko, Exp Mol Med, 2022; Low, mAbs, 2022; Tam, mAbs, 2013, Petkova, Int Immunol, 2006). It is true, however, that in some rare cases administered IgG antibodies can be immunogenic in human FcRn transgenic mice and elicit anti-drug antibodies (ADA) (Roopenian, Protocol, 2016). When this occurs, the beta phase clearance curves are characterized by a biphasic behavior with a rapid drop in antibody concentration starting at approximately 5-6 days post-administration. This problem can then be overcome by using immunodeficient human FcRn transgenic mice (Roopenian, Protocol, 2016; Myzithras, mAbs, 2016).

In our case, we did not measure any rapid drop in plasma clearance. Still, this cannot exclude that some anti-human IgG ADA responses were generated. To address this, we measured both total IgG levels and mouse anti-human IgG levels in plasma samples from the half-life experiments with anti-NIP and anti-SARS-CoV-2 (mAb4) IgG1 WT and REW. Total IgG levels were measured by an anti-human IgG ELISA, while mouse anti-human IgG antibody levels were measured by coating NIP-WT, NIP-REW, mAb4-WT or mAb4-REW in ELISA wells followed by addition of plasma samples from all collected timepoints and detection of bound mouse IgG antibodies. The results showed a monophasic clearance of total IgG and we only measured very minor, or no mouse anti-human IgG antibody responses compared to pre-bleed samples. Anti-human IgG levels were slightly higher in mice receiving NIP WT and mAb4 WT than in mice receiving the REW-containing IgG1 variants. We speculate that it may also be that the administered IVIg could exert a “buffer effect” to shield the test antibodies from any potential ADA mediated clearance as these constitute a minor fraction of the total human IgG levels in the animals.

The new results have been added to Fig. S3h-k, which are described in the result section (line 183-186), method section (line 506-514 and line 601).

6. In Figure S3l, why there is no difference in FcRn binding between WT and REW Fc fusions of HA?

We fully agree with the reviewer that the FcRn binding data presented in Fig. S3l was suboptimal as the data showed only a minor increase in binding of HA-Fc-REW over HA-Fc-WT under acidic pH conditions (6.0). Based on experience, we believe that this is a result of the ELISA set-up used where the monovalent Fc fusions were coated directly in the wells in contrast to the FcRn binding ELISAs where full-length anti-NIP antibodies were captured on their cognate antigen. To correct for this, we have reproduced and purified the Fc fusions and tested them in an established alternative human FcRn binding ELISA. Here,

soluble human FcRn is captured on a human albumin variant engineered for pH independent binding to the receptor (Grevys, Nat Com, 2018). Then, titrated amounts of HA-Fc-WT or HA-Fc-REW are added at either pH 6.0 or 7.4. As a negative control, we included anti-NIP IgG1-H310A, which should not bind the receptor, and binding was detected using an anti-human IgG Fc AP-conjugated antibody. The results clearly show stronger pH dependent human FcRn binding of HA-Fc-REW compared to the WT fusion while anti-NIP IgG1-H310A did not bind at either pH condition.

The new ELISA results have been added to Fig. S3s and the method section updated (line 549-558).

In addition to the reviewer's comments, the prediction of T-cell epitopes using an updated version of NetMHC4.1 has been re-run and updated as described in the materials and method section (line 815-817) and the results have been updated (Table. S11).

Further, Aina Anthi has been included as co-author based on experimental contributions during the review process.

Reviewer #2 (Remarks to the Author):

The manuscript by Foss and colleagues addresses a timely and highly interesting topic. Identifying IgG variants with optimized effector functions and tissue distribution is key to achieve better clinical responses in treating cancer, autoimmunity as well as infectious diseases. The study represents a tour de force, starting with the identification of an IgG variant (REW) which better FcRn binding based on crystal structure data, characterization of the functional properties of this variant (complement activation, cancer cell killing in vitro, etc..), to ultimately testing the enhanced function of this variant in in vivo model systems of influenza infection. In general, the study is performed at very high standards, contains clearly present and interpretable results and demonstrates that the REW IgG variant may have great potential.

Suggestions to further improve the impact of the study:

1. The influenza vaccination study is impressive and suggests that the HA-IgG1REW construct induces a stronger protective immune response. Do the authors have further immunologic data to support this very interesting result (anti-HA antibody levels or enhanced HA specific cytotoxic T cells)?

We thank the reviewer for this question. Based on our *in vitro* and *in vivo* experiments showing enhanced transcellular transport of an REW-containing IgG1 compared to the WT, we were motivated to address the benefit of REW for an Fc-fusion design. To do so, we choose to include a vaccine example where the globular domain of HA from H1N1 was fused to IgG1-Fc WT or REW. The formats were used for intranasal delivery in the presence of adjuvant. In line with improved human FcRn binding of the REW containing variant, we measured a more protective response compared with the WT fusion counterpart upon challenge with a deadly dose of the virus. We agree with the referee that this is a highly interesting finding, which deserves further investigation regarding the mechanism of action and the immune responses induced upon vaccination. While we measured complete protection, there are some limitations to the experimental set-up that are discussed below.

To partly answer the question, we have used the plasma samples collected from the mice to measure the presence of anti-HA IgG antibodies using an established ELISA. The results showed a modest increase in anti-HA specific antibodies in mice vaccinated with HA-Fc-REW compared to HA-Fc-WT while no HA specific antibodies were detected in plasma from mock treated (NaCl) animals. While these data are

indicative, we would like to highlight that the levels of detected antibodies were low, which is in line with the limitation described above related to the fact that human FcRn expressing mice do not rescue endogenous IgG from degradation. As such, while the mice are responding to the vaccine by generating anti-HA IgG antibodies, these are not rescued by FcRn-mediated recycling (short half-life).

The new data has been added to Fig. S3v and described in the result section (line 212-213) as well as in the method section (line 570-571 and 574-583).

As the above biology hampers extensive studies on the material at hands, it will require large set-ups for *in vivo* studies to be run over an extended time to gain an in-depth understanding, which we argue is beyond the scope of the current manuscript focusing on the design of the REW technology and its versatility. However, as we share the enthusiasm by the referee, we have initiated a project where we are exploring this finding further, but then in the context of a new mouse model expressing both human FcRn and human IgG1-Fc (Low, mAbs, 2020). In addition, we are in this context also exploring a new vaccine design that we cannot disclose due to an on-going process related to intellectual property rights. As we have data supporting that this modified strategy is even better, supported by experiments performed in a more optimal double transgenic mouse model, we are not permitted by our animal facility to perform any additional experiments with the setup included in the current manuscript due to ethical reasons (the 3Rs). We sincerely hope that the referee understands this situation and can agree that the on-going work is beyond the main scope of the manuscript.

We have added additional text to the discussion (line 362-367) to emphasize the need for further studies on FcRn-targeted non-invasive delivery of Fc fusions, including subunit vaccines.

2. The authors may consider to include a syngeneic lung tumor model (B16F10 or alike) where cytotoxic antibodies are available. With respect to B16F10 the TA99 antibody could be generated on a human IgG1 format including the REW mutations. Human IgG1 binds well to mouse FcRIV and FcRI and thus should have enhanced activity against lung tumors. This type of experiment is only possible of course if the REW mutant retains binding to mouse FcRs I and IV.

We fully agree that the REW technology should be tested in disease specific animal models to further explore its therapeutic potential against both cancer and infectious diseases. While the B16F10 syngeneic lung tumor model could potentially be used based on the mouse FcγR properties of REW (Fig. A1 below), this model has not been adopted to human FcRn transgenic mice. Regarding cancer, our goal is to address the combined effect of extended plasma half-life, biodistribution and complement activation, but also FcγR mediated effector functions using REW and low fucose REW. Due to FcRn-IgG cross species differences, such experiments must be carried out in mice that express both human FcRn and human FcγRs. Although such mice have been reported (Lee, Nat Com, 2019) they are currently not available to the wider research community. However, we are in the process of exploring the possibility to obtain similar mice expressing both the human FcγRs and human FcRn. It is pivotal that the model is expressing the human form of FcRn, as REW is disrupting pH dependent binding to mouse FcRn (Fig. S3n), which results in a shorter half-life in wild-type mice than the WT counterpart (Fig. S3o).

Meanwhile efforts have been taken to adopt an *in vivo* infection model of *N. gonorrhoeae* (Gulati, PLoS Biol, 2019) to human FcRn transgenic mice in which the combined effect of plasma persistence, biodistribution and complement activity of REW can be addressed. This is also motivated by the fact that REW favorably engages mouse C1q as discussed below. This work is still in process as we are working on transferring the infection protocol between the different mouse strains.

Text has been added to the discussion (line 370-374 and 395-397) to elaborate on the importance of establishing disease specific humanized transgenic models for studies of IgG Fc-engineering strategies involving half-life extension strategies, such as REW.

Figure A1. Mouse Fc γ R binding properties of REW. (a) Illustration of ELISA assay used to measure binding of Fc-engineered anti-NIP IgG1 WT and REW to mouse Fc γ Rs. (b-e) ELISA binding of antigen captured anti-NIP IgG1 WT and REW to mFc γ RI, mFc γ RIIb, mFc γ RIII and mFc γ RIV. Shown as binding relative to anti-NIP IgG1 WT, mean \pm s.d of duplicates.

3. The authors correctly note that the REW variant should not result in spontaneous complement activation in the serum. Thus, it would be great to show that *in vivo* no spontaneous activation of the complement system occurs. As before it would need to be assessed if mouse C1q binds to REW in a comparable fashion to human C1q. If it does, this would be a highly interesting set of data.

We thank the reviewer for this suggestion and fully agree. To address this, we performed an ELISA where anti-NIP IgG1 WT, REW and P329A were captured on the antigen followed by adding mouse C1q, derived from mouse serum, which was detected with an anti-mouse C1q antibody. The results revealed enhanced binding of REW compared with the WT while the negative control P329A did not bind. As such, the favorable on-target C1q binding phenotype of REW is similar for mouse and human C1q.

The new results have been added to Fig. S5e-f and described in the result (line 230) and method sections (line 613-614 and line 617-620).

Further, we agree that it is highly relevant to investigate whether spontaneous activation of the complement system occurs *in vivo*. To address this, we first asked whether anti-NIP IgG1 WT, REW or RGY could form complexes with mouse C1q *in vivo* by injecting the variants into Tg32 human FcRn transgenic mice at a dose of 10 mg/kg (3 mice per group) followed by isolation of plasma samples after 24 hours. We then used an anti-human IgG Fc specific ELISA to determine the levels of the antibody variants in plasma. The results showed slightly higher plasma concentrations of REW compared to WT (as expected in the absence of IVIg) while the RGY variant, which is known to form spontaneous complexes in the absence of antigen (Diebold, Science, 2014), was shown to be cleared more rapidly, present at 10-fold lower concentrations than WT, as determined by ELISA.

To evaluate whether any of the anti-NIP IgG1 variants bound mouse C1q in plasma, we coated ELISA plates with an anti-mouse C1q antibody followed by addition of plasma samples ELISA-normalized for IgG1 concentrations before detection using an anti-human IgG Fc AP-conjugated antibody. The results showed that only the RGY variant formed complexes with mouse C1q in plasma while no binding of WT or REW were detected as the signal was comparable to that of plasma from non-treated mice.

Furthermore, we measured the level of mouse C3a, a known marker for complement activation, in plasma samples ELISA-normalized for IgG1 concentration. The results showed that only RGY gave rise to elevated

mouse C3a levels in mice while WT and REW treated mice showed comparable signals to that of non-treated mice.

The new results have been added to Fig. S5g-l and described in the result (line 243-248) and method (line 634-648) sections.

4. As the REW variant shows increased FcRn binding it would be of great interest to study at which concentration or if at all it behaves as a bona fide FcRn blocking agent. This could be studied by looking at enhanced clearance of IVIg or human serum albumin compared to untreated or efgartigimod (or any other FcRn blocker).

This is indeed a very interesting question as it relates to how FcRn handles its ligands and engineered versions through the cellular pH gradient during the recycling process. Several anti-FcRn blockers are in the pre-clinical or clinical pipeline, and these have distinct differences in their structural architecture and mode of binding, where two of them have been approved for clinical use to treat myasthenia gravis. Their ability to lower systemic endogenous IgG antibody levels relies on how well they block the IgG binding site on FcRn with sufficient affinity throughout the pH gradient to prevent rescue from degradation of endogenous IgG, as reviewed (Peter, *J Allergy Clin Immunol*, 2020; Wolfe, *J Neurol Sci*, 2021; Zhu, *Neural Regen Res*, 2023; Pyzik, *Nat Rev Immunol*, 2023). Notably, the affinity threshold at neutral pH to allow efficient release of an engineered full-length IgG antibody during exocytosis has been estimated to be in the range of 1 μ M (Borrok, *J Biol Chem*, 2015).

While most of the FcRn blockers in the pipeline are built on full-length IgG antibodies with Fab arms blocking the IgG binding site on FcRn at both acidic and neutral pH, the engineered Fc fragment efgartigimod differs as it is an IgG1-derived Fc fragment with five amino acid substitutions (M252Y/S254T/T256E/H433K/N434F) (YTE/KF) at the core Fc binding site for the receptor. This Fc-engineering approach improves human FcRn binding at both acidic and neutral pH but still a roughly 100-fold difference in affinity is measured between the two pH conditions. As such, efgartigimod has a longer plasma half-life than that of the full-length IgG blockers. Importantly, the improvement in binding at both acidic and neutral pH conditions is far from that of REW or similar Fc-engineering strategies where pH dependency is the key to secure efficient cellular rescue by FcRn (Grevys, *Nat Com*, 2018).

In this regard, another highly interesting difference between the Fc-engineered fragment and the full-length IgG1 antibody counterpart is their distinct ability to be taken up by cells and to accumulate intracellularly in FcRn-positive endosomes, where the Fc fragment can engage cell-membrane FcRn more efficiently, and as such, result in enhanced cellular uptake and accumulation. This has also been demonstrated in non-human primates where the Fab arms of a full-length IgG antibody was shown to negatively affect clearance of endogenous IgG compared to that of the YTE/KF-engineered Fc-fragment (Brinkhaus, *Nat Com*, 2022). The effect of the Fab arms on cellular uptake has also been reported by us for wild-type v.s. that of full-length IgG and Fc-fusions (Gjølberg, *Commun Biol*, 2022). These aspects should be carefully taken into consideration when designing new alternatives or formats with the purpose to perform equally well or better than the candidates reported so far.

Before development of such specific FcRn blockers was initiated, it was already known that administration of high doses of pooled human WT IgG (IVIg) can increase degradation of endogenous IgG through competition for FcRn (Li, *J Clin Invest*, 2005; Vaccaro, *Nat Biotechnol*, 2005). Based on this, and the fact that an IgG Fc fragment has increased ability to be taken up by cells, it is expected that both a full-length IgG and an Fc fragment containing the REW amino acid substitutions will have the capacity to block the IgG

binding site on FcRn and accelerate the clearance of endogenous IgG. In addition, we assume that there will also be a difference in their blocking capacity due to the features discussed above. However, the efficacy and dose requirement to achieve potent blocking must be addressed. As REW is in the low binding range at physiological pH (7.4) (SI Text 5 and Fig. S9 in our manuscript) compared to other engineering strategies, it may underperform as an FcRn blocker at clinically relevant doses compared to existing strategies, but the relation between binding kinetics as a function of pH in this regard should be thoroughly investigated by comparing a panel of variants and designs. This is indeed the focus on an on-going project in the lab, which we hope the reviewer agree, is beyond the scope of the current manuscript.

However, what we have addressed in the context of the manuscript is if the given dose of REW-containing full-length IgG1 (5 mg/kg) affects the IVIg levels (500 mg/kg) in the PK experiments included. To address this, we measured the total human IgG levels in the plasma samples from the half-life experiments by established ELISA. The results showed that neither the REW-containing IgG1 with specificity for NIP or SARS-CoV-2 reduced the total plasma levels of total human IgG compared with that of the WT counterparts.

These results have been added to Figure 3e-f and described in the result (line 177-179) and method (line 506-514 and line 601) sections.

In addition to the reviewer's comments, the prediction of T-cell epitopes using an updated version of NetMHC4.1 has been re-run and updated as described in the materials and method section (line 815-817) and the results have been updated (Table. S11).

Further, Aina Anthi has been included as co-author based on experimental contributions during the review process.

REVIEWERS' COMMENTS

Reviewer #1 (Remarks to the Author):

The authors have sufficiently addressed most of the comments and concerns raised during the review process and included in the revised manuscript additional experimental data that support the study conclusions.

In response to the reviewers' comments, the authors have analyzed the antibody titers elicited upon HA-Fc vaccination, showing a very modest, but not significant difference between WT and REW Fc fusions, which is partly attributed to the selected mouse strain and the interspecies differences in FcRn binding of mouse IgGs. However, a more comprehensive immunological analysis is necessary to draw any conclusions and demonstrate the improved immunogenicity of the REW Fc fusion constructs. Since additional experiments to characterize the immune responses upon HA-Fc vaccination are beyond the scope of the current study, it might be preferable to leave out entirely the HA-Fc vaccination study.

At its current form, the presented vaccination experiment with the HA-Fc fusion constructs is still very preliminary, lacks sufficient experimental rigor (unknown how many times it was repeated; currently only used 4-5 mice per group in the challenge experiment), and does not provide adequate insights into the mechanisms that account for the observed effect.

Reviewer #2 (Remarks to the Author):

The authors have addressed all my questions and concerns.

January 20th, 2024

Point-by-point response to reviewer: NCOMMS-23-15179-T.

REVIEWERS' COMMENTS

Reviewer #1 (Remarks to the Author):

The authors have sufficiently addressed most of the comments and concerns raised during the review process and included in the revised manuscript additional experimental data that support the study conclusions.

In response to the reviewers' comments, the authors have analyzed the antibody titers elicited upon HA-Fc vaccination, showing a very modest, but not significant difference between WT and REW Fc fusions, which is partly attributed to the selected mouse strain and the interspecies differences in FcRn binding of mouse IgGs. However, a more comprehensive immunological analysis is necessary to draw any conclusions and demonstrate the improved immunogenicity of the REW Fc fusion constructs. Since additional experiments to characterize the immune responses upon HA-Fc vaccination are beyond the scope of the current study, it might be preferable to leave out entirely the HA-Fc vaccination study.

At its current form, the presented vaccination experiment with the HA-Fc fusion constructs is still very preliminary, lacks sufficient experimental rigor (unknown how many times it was repeated; currently only used 4-5 mice per group in the challenge experiment), and does not provide adequate insights into the mechanisms that account for the observed effect.

In response to the reviewer and editors' (to keep the vaccine data) comments, we have modified the text in the discussion section to highlight the preliminary nature of the vaccine example of the REW technology. This has been done in the context of the cross-species difference in FcRn-IgG binding between mice and man, and where further studies should be conducted in mice than are not only transgenic for human FcRn but also human IgG.

Text in the result section (line 211-214) and the discussion (line 366-372) has been added and is marked in yellow in the manuscript.